# S-STE: Continuous Pruning Function for Efficient 2:4 Sparse Pre-training

**Yuezhou Hu**[1], **Jun Zhu**[1], **Jianfei Chen**[1][†]

[1]Dept. of Comp. Sci. & Tech., Institute for AI, BNRist Center,
Tsinghua-Bosch Joint ML Center, THBI Lab, Tsinghua University.
`huyz21@mails.tsinghua.edu.cn`, `{dcszj,jianfeic}@tsinghua.edu.cn`

## Abstract

Training deep neural networks (DNNs) is costly. Fortunately, Nvidia Ampere and Hopper GPUs can accelerate matrix multiplications twice as fast as a dense equivalent by implementing 2:4 sparsity. However, previous STE-based 2:4 pre-training methods (*e.g.* STE with hard-thresholding, SR-STE) suffer from optimization difficulties because of discontinuous pruning function. In this study, we comprehensively analyse the bottleneck of traditional N:M sparse training and recognize three drawbacks with discontinuity: incorrect descending direction, inability to predict the amount of descent and sparse mask oscillation. In light of this, we propose S-STE, a simple yet powerful 2:4 training method that contains two parts: to continuously project weights to be 2:4 sparse, and to rescale sparse weights with a per-tensor fixed scaling factor. Besides, we adopt minimum-variance unbiased estimation for activation gradient and FP8 quantization for whole process. Results show that our method surpasses previous 2:4 pre-training recipes and is comparable even with full parameter models. Our toolkit is available at `https://github.com/huyz2023/2by4-pretrain`.

## 1 Introduction

Large scale transformers have achieved many impressive results such as chatbots [43], text-to-video generation [27], and robot manipulation [53]. However, the pre-training of these models is extremely expensive, typically requiring thousands of GPUs to train for months [5]. One possible way to accelerate deep learning computation is sparsity. N:M sparsity [31] is a hardware-friendly sparsity pattern, where every group of $M$ dimensions only has $N$ non-zero entries. Nvidia Ampere GPUs can multiply a 2:4 sparse matrix with a dense matrix, twice as fast as multiplying two dense matrices.

While N:M sparsity has been successfully applied to accelerate inference [31, 40, 15, 35, 9], extending the acceleration to pre-training is highly challenging. To accelerate pre-training, the sparse model must be trained from scratch (random initialization), and the network must stay sparse at all training iterations. To meet these requirements, the algorithm should be able to actively explore connectivity patterns within the constrained N:M parameter space. Therefore, popular pruning methods such as single-shot pruning [24], iterative magnitude pruning [13, 29], and RigL [11] cannot be directly applied to this scenario. Moreover, besides forward propagation, the matrix multiplications in back propagation must be sparsified as well, to provide reasonable training speedup.

Methods based on the straight-through estimator (STE) [52, 2] have shown promise towards solving the challenging problem of sparse pre-training. They maintain a dense weight, which is sparsified in each iteration for fast forward&backward computation, and the dense weight is then updated with STE gradients. In this way, connectivity patterns can be learned jointly with weights in an end-to-end

---

[†]Corresponding author.

fashion with stochastic gradient optimizers. SR-STE [52] is such a method to train sparse networks from scratch, with a regularization term to stabilize the training. Several subsequent works [21, 51, 8] further accelerate back propagation with sparse computations, and Hu et al. [20] applied it for pre-training language models. However, these sparse training methods still have an accuracy gap compared to dense training. Moreover, SR-STE introduces a regularization strength hyper-parameter, which is hard to tune. Due to these limitations, N:M sparsity is not yet used to accelerate pre-training.

In this work, we study STE-based pre-training from the optimization perspective. We point out that STE-based pre-training defines a *discontinuous* loss function, which existing optimization theory and algorithms cannot handle. We reveal several intriguing phenomena highlighting the difficulty of discontinuous optimization, including incorrect descending direction, inability to predict the amount of descent, and oscillation. We sidestep the curse of discontinuity by proposing smooth straight-through estimator (S-STE) as a solution. Cruically, S-STE introduces a new pruning function, which uses a continuous projection function to prune weights to be 2:4 sparse, and scales all nonzero elements to minimize the mean-square-error between original dense weight vector and sparse weight vector. The proposed 2:4 soft-thresholding function is *continuous* but can still generate N:M sparse weights at all times. In this way, the objective function is continuous, and gradient-based optimizers can be readily used. Furthermore, S-STE does not introduce any hyper-parameter, so its practical adoption is easier than SR-STE.

We devise comprehensive pre-training experiments on S-STE, including WMT machine translation, GPT-2 pre-training and, DeiT image classification. Results show that our method surpass previous 2:4 pre-training recipes on a wide range of tasks.

## 2 Formulation of sparse pre-training

The training a neural network can be formalized as an optimization problem $\min_{\mathbf{w}} F(\mathbf{w})$, where $\mathbf{w} \in \mathbb{R}^d$ is the parameter and $F$ is a differentiable empirical risk function: $F(\mathbf{w}) = R_n(\mathbf{w}) = \frac{1}{n} \sum_{i=1}^{n} f(\mathbf{w}; \xi_{[i]})$. Here, $f$ is the loss function, $n$ is the size of data set $\mathcal{D} = \{\xi_{[i]}\}_{i=1}^{n}$ and $\xi_{[i]}$ is the $i$-th sample. The optimization can be solved with standard stochastic gradient method (SG) [4]. Suppose the network is initialized with $\mathbf{w}_1$, $\{\alpha_k\}$ is a positive learning rate sequence, and $\xi_{[i_k]}$ is randomly chosen from $\{\xi_{[i]}\}_{i=1}^{n}$. Then, iteratively we have $\mathbf{w}_{k+1} = \mathbf{w}_k - \alpha_k \nabla_{\mathbf{w}_k} f(\mathbf{w}_k; \xi_{[i_k]})$. As we consider pre-training tasks, $\mathbf{w}_1$ is simply a random initialization.

The training of a sparse network involves optimizing the parameter $\mathbf{w}$ in a constrained space $\mathcal{W} \subset \mathbb{R}^d$. For an N:M-sparse network, the parameter can only have $N$ non-zero elements in each contiguous $M$ dimensions.

Alternative to constrained optimization, we can solve the unconstrained problem:

$$\min_{\mathbf{w}} F(\tilde{\mathbf{w}}) \text{ where } \tilde{\mathbf{w}} = S(\mathbf{w}). \tag{1}$$

Here, $S$ is a pruning function which converts a dense weight $\mathbf{w}$ to a sparse weight $\tilde{\mathbf{w}} \in \mathcal{W}$. One common choice is the hard-thresholding pruning function [52, 45]. For every block of four adjacent elements $\mathbf{a} = [a_1, ..., a_M]^\top \in \mathbb{R}^M$ in the weight vector $\mathbf{w}$, the pruning function can be defined as

$$(S_h(\mathbf{a}))_i = \begin{cases} a_i & \text{if } |a_i| \geq t \\ 0 & \text{if } |a_i| < t \end{cases}, \text{ for } i = 1, ..., M, \tag{2}$$

where $t$ is $N$-th largest element in $\mathbf{a}$.[1] This essentially performs magnitude-based pruning, by zeroing out the two smallest elements. The hard thresholding function can also be written as $S_h(\mathbf{a}) = \mathbf{a} \odot m_h(\mathbf{a})$, where $m_h(\mathbf{a})$ is a 0/1 mask vector, with $(m_h(\mathbf{a}))_i = 1$ if $|a_i| > t$.

However, Eq. (1) cannot be directly optimized since the pruning function $S$ is not differentiable. Particularly, the derivative of the hard-thresholding function $S_h$ is undefined on boundary where the second largest and third largest element have the same magnitude. Therefore, straight-through estimator (STE) [52] is for training, by approximating $\nabla_{\mathbf{w}} f \approx \nabla_{\tilde{\mathbf{w}}} f$ and therefore $\partial S_h(\mathbf{a})/\partial \mathbf{a} \approx \mathbf{I}$:

$$\mathbf{w}_{k+1} = \mathbf{w}_k - \alpha_k \nabla_{\tilde{\mathbf{w}}_k} f(\tilde{\mathbf{w}}_k; \xi_{[i_k]}). \tag{3}$$

---

[1] In this paper, when talking about large and small, we refer to the magnitude. For example, "second largest element" means the element with second largest absolute value.

With the pruning function and STE, each iteration of sparse training involves: (1) prune the dense weight to get the sparse weight: $\tilde{\mathbf{w}} = S(\mathbf{w})$; (2) compute the loss and gradient with the *sparse* weight; and (3) update the *dense* weight with the gradient. Among these, step 2 is most time-consuming, and it can be accelerated with sparse tensor cores given $\tilde{\mathbf{w}}$ is N:M-sparse. Next, we will focus on the optimization aspects of sparse training and defer the discussion of computation details to Sec. 4.

## 3 The curse of discontinuity

Classical stochastic optimization theory [4] guarantees the convergence for nonconvex and *smooth* (i.e., differentiable with Lipschitz continuous gradients) objective $F$. It can be also extended to handle non-differentiable functions such as ReLU [25]. The real problem of STE-based sparse training is the *discontinuity* of the pruning function $S_h$, as visualized in Fig. 2. For a discontinuous function, an arbitrarily small change in input $\mathbf{a}$ can cause an unbounded change of the output $S_h(\mathbf{a})$. Such discontinuity appears on the boundary when the $N$-th and $N+1$-th largest elements have same magnitude. For example, for a 1:2-sparse pruning function, $S_h(1, 0.999) = (1, 0)$, but $S_h(0.999, 1) = (0, 1)$, and the boundary is the line $a_1 = a_2$.

When $S_h$ is discontinuous, the parameter space $\mathbb{R}^d$ can be partitioned into regions $\{\mathcal{W}_\mathbf{m} | \mathbf{m} \in \mathcal{M}\}$, where $\mathcal{M} \subset \{0,1\}^d$ is the space of 0/1 masks with N:M pattern, and all the parameters in each region $\mathbf{w} \in \mathcal{W}_\mathbf{m}$ have the same mask $m_h(\mathbf{w}) = \mathbf{m}$. The loss landscape $F(S_h(\mathbf{w})) = F(m_h(\mathbf{w}) \odot \mathbf{w})$ is continuous and differentiable within each region, where gradient-based algorithms can work well. However, when the optimization trajectory crosses the boundary: $m_h(\mathbf{w}_{k+1}) \neq m_h(\mathbf{w}_k)$, the behavior is unpredictable. We highlight several intriguing phenomena observed in optimizing such discontinuous objective. We study these phenomena in both a toy problem and real neural networks.

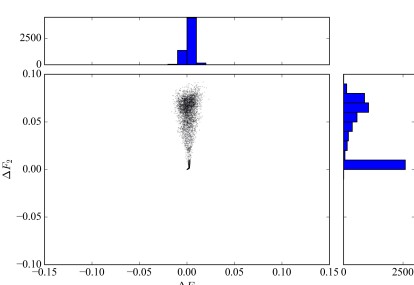

Figure 1: Scatter plot of $\Delta F_1$ with $\Delta F_2$ and their distributions on GPT-2 small 124M for iteration $k \in [1, 6000]$.

### 3.1 Phenomenon 1: incorrect descending direction

Here, we run a gradient descent algorithm (without stochasticity) on a small dataset. For a dense model where $F$ is differentiable, with Taylor's formula we should have

$$F(\mathbf{w}_k) - F(\mathbf{w}_{k+1}) \approx (\nabla_{\mathbf{w}_k} F(\mathbf{w}_k))^\top (\mathbf{w}_k - \mathbf{w}_{k+1}) = \alpha_k \|\nabla_{\mathbf{w}_k} F(\mathbf{w}_k)\|^2 \geq 0. \quad (4)$$

That is, the objective function will monotonically decrease in each iteration once the learning rate $\alpha_k$ is sufficiently small. However, it is not the case for sparse training. In Fig. 2(d), we measure the distribution of the amount of descent (AoD) $\Delta F_k := F(\mathbf{w}_k) - F(\mathbf{w}_{k+1})$ for training a GPT-2 large 774M model with Eq. (2, 3), across each iteration $k$. The results clearly shows that the objective frequently fails to descent.

We can take a closer look to the weight and mask sequence $(\mathbf{w}_k, m_h(\mathbf{w}_k))$ generated by the training algorithm. We compare the following two quantities: the AoD by updating both weight and mask $\Delta F_1 = F(\mathbf{w}_k \odot \mathbf{m}_k) - F(\mathbf{w}_{k+1} \odot \mathbf{m}_{k+1})$ and the AoD by only updating the weight $\Delta F_2 = F(\mathbf{w}_k \odot \mathbf{m}_k) - F(\mathbf{w}_{k+1} \odot \mathbf{m}_k)$. In Fig. 1, we can observe $\Delta F_2$ is mostly positive due to the piecewise continuity of $F$. However, $\Delta F_1$ is frequently negative and very often even smaller than $\Delta F_2$ (updating mask is worse than not updating). This indicates that the main problem is the discontinuity make it hard to estimate the correct descending direction of $\mathbf{m}$.

### 3.2 Phenomenon 2: inability to predict the amount of descent

Besides making mistakes in finding the correct descending direction, algorithms do not know that they make a mistake, in the sense that they fail to predict the AoD at each step. From Eq. (4), we should have $F(\mathbf{w}_k) - F(\mathbf{w}_{k+1}) \approx (\nabla_{\mathbf{w}_k} F(\mathbf{w}_k))^\top (\mathbf{w}_k - \mathbf{w}_{k+1})$, where the left hand side is the *actual*

AoD, and the right hand side is the *predicted* AoD. We plot the actual AoD against predicted AoD for dense (Fig. 2(a)) and sparse training (Fig. 2(b)). While for dense training, the two quantities closely matches, for hard-thresholding the actual AoD is often lower for the predicted AoD, particularly when the predicted AoD is large. To understand this, note that Eq. (4) only holds for $\mathbf{w} \in \mathcal{W}_{m_h(\mathbf{w})}$. Once $\mathbf{w}_{k+1} - \mathbf{w}_k$ is large enough that $m_h(\mathbf{w}_{k+1}) \neq m_h(\mathbf{w}_k)$, the function crosses a border of $S_h$, and $F$ will have a sudden change which is unpredictable by the gradient.

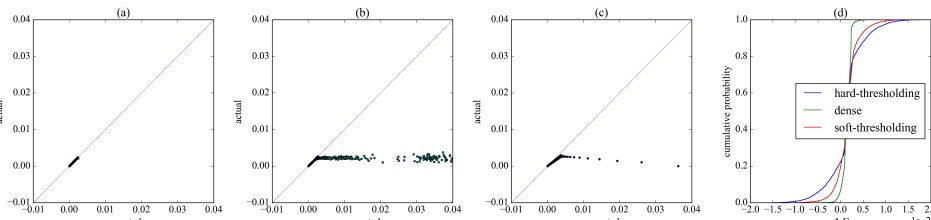

Figure 2: (a)-(c) shows scatter plots of the predicted and actual loss reduction of dense, hard-thresholding and S-STE with GPT-2 large 774M model for iteration $k \in [1, 3000]$. The diagonal line is for reference. (d) shows empirical cumulative distribution of their actual AoD for $k \in [1, 6000]$.

## 3.3 Phenomenon 3: oscillation

Oscillation is probably the most significant problem in STE-based sparse training. Here, we revisit existing discussions about oscillation [52, 28, 20], and then illustrate this issue using a toy example.

**Flip rate** Flip rate is a simple metric to measure the stability of sparse training [20]: $r_k = \|m_h(\mathbf{w}_k) \oplus m_h(\mathbf{w}_{k-1})\|_1 / d$, where $\oplus$ indicates XOR operation. As Hu et al. [20] points out, taking the flip rate of the dense model as standard, they observe larger flip rate of hard-thresholding: when training transformers, the flip rate can stay at 6% in the entire training process. However, a healthy training process should have a large flip rate in the early stage to explore connectivity patterns, and the flip rate should decrease to zero in later stage for the optimization to converge. Hu et al. [20] describe this phenomenon as "flip rate explosion", which is harmful to sparse training.

**An exemplar toy problem** Modern deep neuron networks have billions of parameters and is not strictly convex. These non-ideal conditions make our analysis more difficult with sparse weights. To analyze the characteristics of STE and hard-thresholding on the smallest problem, we devise a simple toy problem that contains two parameters: $\min_{w_1, w_2} g(w_1, w_2) = (w_1 - w_2)^2$. This may differ from the real DNN optimization problem, but can help us understand what happens in the process. We are going to show that while using a feasible $\alpha_k$ that can make the dense model converge to global minima, STE with hard-thresholding fails to converge and it oscillates back and forth.

First, for the dense model, the global minima lies on the line $w_1 = w_2$. Suppose we start from $\mathbf{w}_1 = [0.2, 0.1]^\top$, by taking $\alpha_k = 0.25$ we can reach global minima in one step. On the other hand, if we are in 1:2 sparse situation, the global minima should be the point $w_1 = w_2 = 0$. By starting from $\mathbf{w}_1 = [0.2, 0.1]^\top$ and taking $\alpha_k = 0.25$, we invariably jumps between $\mathbf{w}_{2t+1} = [0.2, 0.1]^\top$ and $\mathbf{w}_{2t} = [0.1, 0.2]^\top$, and $g$ never decreases.

High flip rate is harmful, because there are frequent changes on the connection of neurons, which means that a number of previous optimization steps on the neuron is deprecated. That is fatal at the end of training [20]. The reason of high flip rate on hard-thresholding can be explained by discontinuity: as there are no gentle transitions on both sides of the border, the gradient on the boundary is inaccurate and is unable to indicate the right descending direction. This misalignment is easy to make the tuple $\mathbf{a}$ to oscillate back and forth near the boundary, and cause extremely higher flip rate than the dense model.

## 3.4 Overcoming the curse of discontinuity

One way to mitigate discontinuity is sparse-refined straight-through estimator (SR-STE), which adds a sparse-refined regularization on the gradients [52]: $\min_{\mathbf{w}} F(\tilde{\mathbf{w}}) + \frac{\lambda_W}{2} \|\mathbf{w} \odot \overline{m(\mathbf{w})}\|_2^2$. While

SR-STE works on a wide range of tasks and optimization algorithms [52, 20], it still has some issues. First, the performance is quite sensitive to the hyper-parameter $\lambda_W$. Second, the new regularization term leads to a competition between loss function and sparse regularization. Finally, the loss function is still discontinuous unless $\lambda_W \rightarrow \infty$.

From the above analysis, discontinuity causes optimization problems. It would be ideal to have a *continuous* pruning function, yet the iterates $(\tilde{\mathbf{w}}_k)$ still need to be sparse during the entire training process.

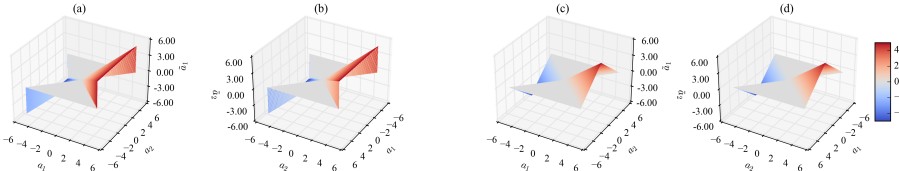

Figure 3: Pruning function of hard-thresholding and soft-thresholding for 1:2-sparsity. (a)(b) show the outputs of hard-thresholding, and (c)(d) show that of soft-thresholding. A sudden jump exists in hard-thresholding if $|a_1| = |a_2|$, while soft-thresholding is continuous in the domain.

## 4 Methodology

In this section we propose a training algorithm (smooth straight-through estimator, S-STE) that contains two main parts combined with STE: 2:4 specific soft-thresholding, and fixed weight rescaling . They together work as the sparsifying function described in Sec. 2: $\tilde{\mathbf{w}} = S(\mathbf{w}) = \beta S_{soft}(\mathbf{w})$. Results in Fig. 2(c)(d) and 4(d) show that S-STE successfully overcome the three curses of discontinuity. Notably, flip rate curves of S-STE are surprisingly consistent with their dense counterparts, indicating that S-STE is more natural and feasible than SR-STE.

Table 1: Validation loss and test accuracy of S-STE with different $\gamma$ on Transformer-base.

| $\gamma$ | Val loss | Test BLEU |
|---|---|---|
| 0 | 4.007 | 26.30 |
| 0.33 | 4.014 | 26.01 |
| 0.67 | 4.015 | 26.16 |
| 1 | 4.072 | 25.63 |

### 4.1 2:4 specific soft-thresholding $S_{soft}$

**Motivation for the design**  As discussed in Sec. 3, hard-thresholding suffer from the discontinuous problem near the boundary of taking a flip. When input vector changes continuously across the border, two of the four elements simultaneously jump between zeroes and none-zero values. In a continuous pruning function, we need to overcome this drawback and keep these two elements zero on both sides of the border. This means when a flip happens in a four-element block, at lease three of the target elements should be zeroed out simultaneously (except the largest one).

With the above analysis, we modify soft-thresholding function for traditional pruning in Vanderschueren and Vleeschouwer [45] as our 2:4 specific soft-thresholding. Given a vector $\mathbf{a} = [a_1, a_2, a_3, a_4]^\top \in \mathbb{R}^4$, S-STE picks out the largest two elements and meanwhile, subtracts the third largest element from weight magnitudes. Assume, without loss of generality, that $[t_1, t_2, t_3, t_4]$ is an rearrangement of $\mathbf{a}^\top$, $s.t.|t_1| \leq |t_2| \leq |t_3| \leq |t_4|$. Then, the pruning function can be defined as

$$(S_{soft}(\mathbf{a}))_i = \begin{cases} a_i - t & \text{if } a_i \in [t, +\infty) \\ 0 & \text{if } a_i \in (-t, t) \\ a_i + t & \text{if } a_i \in (-\infty, -t] \end{cases}, \text{ where } t = |t_2|. \tag{5}$$

The plots of soft-thresholding is drawn in Fig. 3, showing $S_{soft}$ is continuous everywhere. Note that although we define $S_{soft}$ by a block $\mathbf{a} \in \mathbb{R}^4$, $S_{soft}$ can be extended to arbitrary $\mathbf{a} \in \mathbb{R}^{4t}$ for $t \geq 1$, by doing block-wise pruning.

**Theorem 4.1.** *$S_{soft}(\mathbf{a})$ is a continuous projection for $\mathbf{a} \in \mathbb{R}^d$.*

A detailed discussion of the proof can be found in Appendix A.1.

**Choosing optimal threshold** Theoretically, any real number in $[|t_2|, |t_3|]$ can be used as a feasible threshold. This gives us infinite options and we describe it with an interpolation as $t = \gamma|t_2| + (1 - \gamma)|t_3|$ with $\gamma \in [0, 1]$. The larger $\gamma$ is, the closer $t$ is to $|t_3|$, and the smaller $\|S_{soft}(\mathbf{a})\|$ is. This may affect model's capacity. In order to maximize the retention of information, using a small $\gamma$ is necessary. In our method we propose to set $\gamma = 0$. Experimental results in Table 1 also show that the network has the best accuracy when $\gamma = 0$, *i.e.*, $t = |t_2|$.

## 4.2 Fixed weight rescaling $\beta$

Because $S_{soft}$ reduce the total magnitude of $\mathbf{w}$, it is not a close simulation of dense weights. Like Vanderschueren and Vleeschouwer [45], we scale up $S_{soft}(\mathbf{w})$ in our method as $S(\mathbf{w}) = \beta S_{soft}(\mathbf{w})$, but we modify weight rescaling in their study to adapt to our approach. First, we use a *per-tensor scale $\beta$* rather than a per-channel $\beta$ for simplicity. Besides, two important improvements are made: to compute scale factor only at the beginning of training, rather than to dynamically update scale factor during training, and to minimize the mean-square-error (MSE) between original dense weights and sparse weights, rather than to keep the total magnitude of weights unchanged.

**Freezing scaling factor** As Vanderschueren and Vleeschouwer [45] use a dynamic $\beta$ for every iteration, we argue that this doesn't align with our approach. We explain our solutions in two parts.

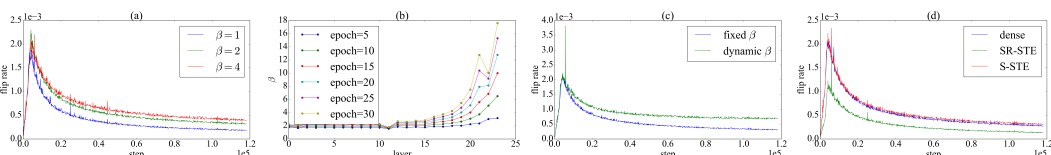

Figure 4: (a) Flip rate curve over the training process with different $\beta$ on Transformer-base. (b) Dynamically recalculated $\beta$ at each layer on different epochs. Results show that frequently updating $\beta$ will cause it to be unexpectedly large. (c) Flip rate curve over the training process with fixed and dynamic $\beta$ on Transformer-base. (d) Flip rate of dense, SR-STE and S-STE algorithm on Transformer-base.

First, we find it interesting that $\beta$ have a subtle correlation with flip rate: *in a sparse model, larger $\beta$ usually results in higher flip rate.* The reason can be explained by the accuracy of gradients. As we use STE in the backward pass, the approximation $\nabla_{\mathbf{w}} f \approx \nabla_{\tilde{\mathbf{w}}} f$ is valid when $\tilde{\mathbf{w}}$ and $\mathbf{w}$ are close enough. However, if scale is too large then optimal, $\mathbf{w}$ and $\tilde{\mathbf{w}}$ are too far apart to guarantee this. Such a mismatch leads to incorrectness in descending directions, and thus unstability in optimization increase flip rate; see Fig. 4(a).

Second, we argue that dynamically computing scaling factor for each iteration leads to high flip rate in our training process. Fig. 4(b) shows the results dynamically changing $\beta$ will make it increase with iterations, especially for later layers. Fig. 4(c) shows flip rate of this network, which has a significantly higher tail than the dense one. Considering high flip rate is harmful, we propose to compute scaling factor $\beta$ only in the first iteration. After that, we use the same $\beta$ in the rest of the training. Fig. 4(d) shows the flip rate of our fixed scaling S-STE, which perfectly aligns with the dense one.

**Minimizing MSE** Vanderschueren and Vleeschouwer [45] choose to scale up $S_{soft}(\mathbf{w})$ to have the same L1-norm as $\mathbf{w}$: $\beta = \|\mathbf{w}\|_1 / \|S_{soft}(\mathbf{w})\|_1$. However, we choose to minimize the MSE gap of $S_{soft}(\mathbf{w})$ and $\mathbf{w}$. As [8] point out, sparsifying weights in the forward pass should minimize MSE rather than an unbiased estimation. In our method, to determine an optimal scale $\beta$, we need to minimize

$$\text{MSE} = \|\mathbf{w} - \beta S_{soft}(\mathbf{w})\|^2 = \|\mathbf{w}\|^2 - 2\mathbf{w}^\top S_{soft}(\mathbf{w})\beta + \|S_{soft}(\mathbf{w})\|^2 \beta^2. \quad (6)$$

Rearrange the terms and taking partial derivative of $\beta$, we choose $\beta = \mathbf{w}^\top S_{soft}(\mathbf{w}) / \|S_{soft}(\mathbf{w})\|^2$. The comparison between no scaling, our minimizing MSE and keeping L1-norm can be found in Table 2. Result show that our method yields the best results in practice.

Table 2: Experimental result of different $\beta$ on Transformer-base.

| $\beta$ Recipe | Test BLEU | Val loss | Avg epoch loss |
|---|---|---|---|
| No scaling | 25.28 | 4.044 | 4.670 |
| Keeping L1-norm same [45] | 25.85 | 4.019 | 4.627 |
| **Minimizing MSE (S-STE)** | **26.3** | **4.007** | **4.605** |

Table 3: Results of different MVUE strategies on GPT-2 774M with 4000 steps. Sparsifying $S(\mathbf{W})^\top$ introduces huge loss of accuracy while sparsifying $\nabla_{\mathbf{Z}}^\top$ is acceptable with little loss.

| S-STE | MVUE($S(\mathbf{W})^\top$) | MVUE($\nabla_{\mathbf{Z}}^\top$) | comment | loss |
|---|---|---|---|---|
| - | ✗ | ✗ | dense | 3.3948 |
| - | ✗ | ✗ | SR-STE | 3.4739 |
| ✓ | ✗ | ✗ | | 3.4333 |
| ✓ | ✓ | ✗ | | 3.4644 |
| ✓ | ✓ | ✓ | | 3.4773 |
| ✓ | ✗ | ✓ | | **3.4480** |

## 5 Other implementation skills

### 5.1 Minimum-variance unbiased estimation

To accelerate the backward propagation, Chmiel et al. [8] suggest using a minimum-variance unbiased estimator (MVUE). For every linear layer $\mathbf{Z}_l = \mathbf{X}_l S(\mathbf{W}_l)^\top$, there are two matrix multiplications of the backward pass in total: $\nabla_{\mathbf{X}_l} = \nabla_{\mathbf{Z}_l} S(\mathbf{W}_l)$ and $\nabla_{\mathbf{W}_l} = \nabla_{\mathbf{Z}_l}^\top \mathbf{X}_l$, where $\mathbf{X}_l$ is the input of the $l$-th layer, $\mathbf{W}_l$ and $\mathbf{Z}_l$ are the weight matrix and output activation. We conduct MVUE on both two matrix multiplications and compare their results: $\nabla_{\mathbf{X}_l} = \nabla_{\mathbf{Z}_l} \text{MVUE}(S(\mathbf{W}_l)^\top)^\top$ and $\nabla_{\mathbf{W}_l} = \text{MVUE}(\nabla_{\mathbf{Z}_l}^\top)\mathbf{X}_l$. Specifically, we choose $S(\mathbf{W}_l)$ and $\nabla_{\mathbf{Z}_l}$ because they both have built-in sparsity [26]. However, we only choose to sparsify the latter one. Firstly, it is proven by Hu et al. [20] and Chmiel et al. [8] that minimum loss of accuracy is guaranteed for MVUE on $\nabla_{\mathbf{Z}_l}$. Secondly, using MVUE on $S(\mathbf{W}_l)$ will make errors accumulate along the back propagation, and results in large standard deviation of gradient for the first few layers. Besides, results in Table 3 also show minimum loss of accuracy of MVUE on $\nabla_{\mathbf{Z}_l}$ while obvious accuracy loss on $S(\mathbf{W}_l)$. Thus, we choose to sparsify only $\nabla_{\mathbf{Z}_l}$ in the backward pass.

### 5.2 FP8 training

To further accelerate pre-training of networks, we utilize popular FP8 workflow in training. Similar to Transformer Engine [2], we use FP8 e3m4 in forward pass and e5m2 in backward pass. Besides, we use per-tensor rescaling before casting to FP8 formats.

**Theoretical acceleration of S-STE** While 2:4 sparsity can accelerate GEMMs up to 2x faster, FP8 quantization can accelerate an additional 2x on this basis. Thus, the three GEMMs in Sec. 5.1 can be 4x, 2x, 4x faster. To sum up, we have theoretically 3x faster in forward and backward pass.

## 6 Experiments

We validate the feasibility of our proposed method S-STE on machine translation (Transformer [46]), image classification (DeiT [42]) and generative large language models (GPT-2 [38] series). For all models, we replace the two linear layers in the feed forward network of each transformer block with

---

[2]`https://github.com/NVIDIA/TransformerEngine`

S-STE. We keep the rest of the networks, the optimization algorithms as well as all hyperparameters the same as their dense counterparts.

For Transformer, we train Transformer-base models on WMT 14 En-De dataset [3] with fairseq [34] codebase and evaluate it with BLEU [36] scores. For DeiT, we pre-train Deit-small model for ImageNet-1K [10] classification task. For GPT-2, we pre-train GPT-2 124M, 350M and 774M models on OpenWebText [16] and evaluate it on GLUE [47] and SQuAD [39] benchmarks. We also compare our method with state-of-the-art 2:4 training methods (SR-STE [52], Bi-Mask [51] and STEP [28]). The pre-training and evaluation scripts are publicly available at `https://github.com/thu-ml/2by4-pretrain-acc-examples`.

**Machine translation** We first apply S-STE to train a 12-layer Transformer-base and compare it with SR-STE and STEP. Note that we use fairseq codebase with SacreBleu metric, whose baseline should be 26.5 (the result of our reproduction is 26.42). The results are shown in Table 4. Compared with SR-STE, our method improves by 0.3 and 0.5 on test set and validation set respectively, which is the closest to baseline. Besides, we improve by 0.6 compared to STEP on test set.

Table 4: Experimental Results for Transformer-base on En-De dataset.

| Method | Avg epoch loss | Test BLEU | Val BLEU | Val loss |
|---|---|---|---|---|
| Dense | 4.555 | 26.42 | 26.49 | 3.977 |
| SR-STE | 4.61 | 25.84 | 26.08 | 4.023 |
| STEP | 4.682 | 25.52 | 26.01 | 4.085 |
| **S-STE** | **4.617** | **26.11** | **26.53** | **4.011** |

Table 5: Experimental Results for DeiT-small on ImageNet-1k. The Bi-Mask and SR-STE results are from [51].

| Model | Method | Test acc1 | Test acc5 |
|---|---|---|---|
| DeiT-tiny | Dense | 72.2 | 91.1 |
| | SR-STE [51] | 67.8 | 88.6 |
| | **S-STE** | **68.5** | **88.9** |
| DeiT-small | Dense | 79.9 | 95 |
| | SR-STE [51] | 75.7 | - |
| | Bi-Mask [51] | 77.6 | - |
| | **S-STE** | **78.5** | **94.4** |

**Image classification** We further investigate the effectiveness of S-STE to train DeiT-tiny and DeiT-small on ImageNet-1k; see Table 5. Results show S-STE also achieve the best performance among different methods, with only has 1.4% degradation from the dense model. Notably, S-STE surpasses SOTA 2:4 training method Bi-Mask on this task (0.9% top1 accuracy improvement) and popular SR-STE method (2.8% top1 accuracy improvement).

**Generative language models** We compare S-STE with dense, normal SR-STE and SR-STE with dense fine-tuning [20] (SR-STE+DF) models on GLUE and SQuAD tasks. The SR-STE+DF models first use SR-STE to train a 2:4 sparse model, and switch to dense training for the last 1/6 iters of pre-training (which stands for "dense fine-tune"). In downstream tasks it also use dense parameters to make predictions, similar to dense models. Results in Table 6 and 9 show that S-STE completely surpasses SR-STE on both tasks. Even for SR-STE+DF models, S-STE still have an advantage, with an improvement of 1.5 on GLUE average and 1.2/0.9 on SQuAD for GPT-2 774M.

**Fine-tuning** We illustrate the viability of S-STE for fine-tuning a pre-trained model, presenting a coherent workflow of accelerating both training and inference (dense fine-tuning cannot produce a sparse model for inference acceleration); see Table 7, 6.

Table 6: SQuAD and GLUE scores of different sizes and pre-training methods on GPT-2. We use 2:4 sparse weights to evaluate S-STE model, while dense parameters to evaluate the rest. Of note, SR-STE denotes the original SR-STE workflow (without backward MVUE), and "T-SR-STE+DF" denotes the combination of transposable SR-STE & backward MVUE & sparse-dense training workflow, proposed by Hu et al. [20]. S-STE settings here include backward MVUE & FP8 training.

| Params | Pre-train | Fine-tune | Pre-train val loss | SQuAD | | GLUE@Avg |
|--------|-----------|-----------|--------------------|-------|------|----------|
| | | | | EM | F1 | |
| 124M | Dense | Dense | 2.907 | 67.6 | 78.8 | $73.9 \pm 1.1$ |
| | T-SR-STE+DF [20] | Dense | 2.952 | 67.5 | 78.5 | $74.3 \pm 0.5$ |
| | T-SR-STE | Dense | 3.076 | 66.3 | 77.2 | $72.6 \pm 0.2$ |
| | SR-STE | Dense | 2.982 | 66.2 | 77.5 | $73.8 \pm 0.3$ |
| | **S-STE** | **S-STE** | **2.984** | **68** | **78.8** | **$74.1 \pm 0.4$** |
| 350M | Dense | Dense | 2.618 | 73.2 | 83.6 | $76.3 \pm 0.1$ |
| | T-SR-STE+DF [20] | Dense | 2.688 | 71.9 | 82.4 | $77.1 \pm 0.2$ |
| | T-SR-STE | Dense | 2.718 | 72.3 | 82.6 | $76.3 \pm 0.4$ |
| | SR-STE | Dense | 2.690 | 72.0 | 82.4 | $76.8 \pm 0.4$ |
| | **S-STE** | **S-STE** | **2.713** | **72.2** | **82.7** | **$76.9 \pm 0.6$** |
| 774M | Dense | Dense | 2.493 | 74.3 | 84.9 | $76.2 \pm 0.4$ |
| | T-SR-STE+DF [20] | Dense | 2.564 | 74.3 | 84.6 | $77.1 \pm 0.4$ |
| | **S-STE** | **S-STE** | **2.547** | **75.5** | **85.5** | **$78.6 \pm 0.8$** |

Table 7: Different fine-tuning results on GLUE and SQuAD.

| Model | Downstream task | Pre-train | Fine-tune | Avg score |
|-------|-----------------|-----------|-----------|-----------|
| GPT-2 124M | GLUE | S-STE | Hard-thresholding | $73.9 \pm 0.6$ |
| | **GLUE** | **S-STE** | **S-STE** | **$74.1 \pm 0.4$** |
| | SQuAD | S-STE | Hard-thresholding | $67.6/78.6$ |
| | **SQuAD** | **S-STE** | **S-STE** | **$68/78.8$** |

**Ablation study**   In this part, We explore the effectiveness of S-STE, MVUE, and FP8 separately. We pre-train DeiT-small model on ImageNet-1K dataset for image classification. Combinations of these partitions in Table 8 show that: 1) FP8 training has little affect on pre-training accuracy (0.2% of acc1); 2) MVUE leads to minimal loss of performance (0.1% of acc1).

Table 8: Experimental result of S-STE (soft-thresholding and weight rescaling), MVUE and FP8 training with DeiT-small on ImageNet-1K.

| Soft-thresholding | Weight rescaling | MVUE($\nabla_{\mathbf{z}}^{\top}$) | FP8 | Comment | Test acc1 | Test acc5 |
|-------------------|------------------|-----------------------|-----|---------|-----------|-----------|
| - | - | ✗ | ✗ | Dense | 79.9 | 95 |
| - | - | ✗ | ✓ | Dense; FP8 | 79.7 | 94.9 |
| ✗ | ✗ | ✗ | ✗ | Hard-thresholding | 77.7 | 93.9 |
| ✓ | ✓ | ✗ | ✗ | | 78.8 | 94.6 |
| ✓ | ✗ | ✗ | ✗ | | 78.9 | 94.7 |
| ✓ | ✓ | ✗ | ✓ | | 78.6 | 94.4 |
| ✓ | ✓ | ✓ | ✗ | | 78.9 | 94.6 |
| ✓ | ✗ | ✓ | ✗ | | 78.2 | 94.2 |
| ✓ | ✓ | ✓ | ✓ | | **78.5** | **94.4** |

**Acceleration** For acceleration, we measure the acceleration ratio of a typical GPT-2 model using implementation from Hu et al. [20]. Note that on H100 GPUs, FP8 2:4-spMM kernel turns out to be unsatisfying; see Appendix A.4. Consequently, we fall back to use RTX3090 GPUs with FP16 training. For inference, we achieve 1.53x speedup with FFN layer and 1.23x speedup with the network; for pre-training, we achieve 1.32x speedup for FFN layer and 1.18x speedup for the network (Appendix A.3).

# 7 Related work

**Unstructured pruning and coarse-grained structured pruning** Pruning is to remove redundant weights from the dense model. Traditional one-shot pruning methods [17, 18, 12, 14, 31, 24] and dynamic sparse training methods [11, 6, 7, 50] mostly target on unstructured sparsity. While most of them have acceleration effect on CPUs, it is hard for these methods to work well on modern GPUs. Coarse-grained structured sparsity [49, 23, 19, 22] takes effect to acceleration, but since they often remove a whole chennel or a block, loss of information is non-negligible.

**Fine-grained N:M sparsity for inference and pre-training** Among all pruning techniques for pre-training, N:M sparsity is a promising approach towards accelerating large models, which is also known as fine-grained structured sparsity. Nvidia demonstrates 2x theoretical speedup on its Ampere GPUs with 2:4 sparsity for post-training [31] and inference [40, 15, 35, 9]. To leverage this property to accelerate pre-training, a number of approaches and their improvements are proposed [52, 28, 51, 1, 8, 21, 20]. However, all these methods are based on a discontinuous pruning function that is hard to optimize and results in unsatisfactory accuracy, which we will discuss in this study.

**FP8 quantization** While 16-bit float tensors are widely used in pre-training, FP8 – where float numbers stored in 8 bits – is a popular quantization methods which theoretically accelerates GEMMs up to 4x faster than its fp32 counterparts and 2x faster than its FP16/BF16 counterparts [30, 37, 44, 48, 32]. With e3m4 data format used in forward and e5m2 format [41] in backward, pre-trained models can achieve minimum loss of accuracy while greatly boosting the efficiency of training.

# 8 Conclusions and future work

In this study we discuss the importance of pruning continuity in effective 2:4 sparse pre-training. We analyse the drawback of traditional hard-thresholding pruning function and its variation (SR-STE) and argue that the main limits being discontinuity. Based on our analysis and soft-thresholding for channel pruning, we propose S-STE, which prunes weights in a continuous manner. Experiments show that our method surpasses previous state-of-the-art methods on a wide range of tasks.

Our proposed S-STE approach primarily targets linear layers within FFN networks. Nevertheless, QKV projection layers necessitate further exploration to devise an effective dynamic sparse training strategy that harmonizes with attention mechanisms. Furthermore, our current choice of continuous pruning function represents only one possible solution; alternative, smoother pruning functions may be necessary to achieve improved continuity and mitigate potential discontinuities.

## Acknowledgments and Disclosure of Funding

We would like to thank Ziteng Wang, Bingrui Li, Haocheng Xi, Kang Zhao and Jintao Zhang, Brian Chmiel and Daniel Soudry for valuable discussions. This work was supported by the National Key Research and Development Program of China (No. 2021ZD0110502) and NSFC Project (Nos. 62376131). J.Z is also supported by the XPlorer Prize.

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

# A  Appendix / supplemental material

## A.1  Proof of Theorem 4.1

*Proof.* We prove this by demonstrating $S_{soft}$ is continuous on every 4-element block. Of note, suppose $\mathbf{a} = [a_1, a_2, a_3, a_4]^\top$. Assume, without loss of generality, that $|a_1| \leq |a_2| \leq |a_3| \leq |a_4|$. Our goal is to prove $\forall \epsilon > 0, \exists \delta > 0, s.t.$ when $|a_1' - a_1| < \delta, |a_2' - a_2| < \delta, |a_3' - a_3| < \delta$ and $|a_4' - a_4| < \delta, |(S_{soft}(\mathbf{a}'))_i - a_i| < \epsilon$, where $\mathbf{a}' = [a_1', a_2', a_3', a_4']^\top$.

1) We start from the simplest case where $|a_1| < |a_2| < |a_3| < |a_4|$. Then we have

$$S_{soft} = [0, 0, \text{sign}(a_3)(|a_3| - |a_2|), \text{sign}(a_4)(|a_4| - |a_2|)].$$

This order holds when

$$\delta < \frac{1}{2} \min\{(|a_2| - |a_1|), |a_3| - |a_2|), |a_4| - |a_3|)\}.$$

Thus,

$$S_{soft}(\mathbf{a}') = [0, 0, \text{sign}(a_3')(|a_3'| - |a_2'|), \text{sign}(a_4')(|a_4'| - |a_2'|)].$$

The signs of $\mathbf{a}$ is unchanged when

$$\delta < \min\{|a_1|, |a_2|, |a_3|, |a_4|\}.$$

We have

$$\text{sign}(a_4')(|a_4'| - |a_2'|) - \text{sign}(a_4)(|a_4| - |a_2|)$$
$$\leq |\,|a_4'| - |a_2'| - |a_4| + |a_2|\,|$$
$$\leq |\,|a_4'| - |a_4|\,| + |\,|a_2'| - |a_2|\,|$$
$$\leq 2\delta$$

Take $\delta \leq \frac{1}{2}\epsilon$ and this is done. It is similar to prove that $\text{sign}(a_3')(|a_3'| - |a_2'|) - \text{sign}(a_3)(|a_3| - |a_2|) \leq \epsilon$ using the same method.

2) We then consider the cases where there are two equivalents in $\mathbf{a}$. If $|a_1| = |a_2| < |a_3| < |a_4|$ or $|a_1| < |a_2| < |a_3| = |a_4|$, the proof should follow 1) as no flip happens. Thus we only consider the situation where $|a_1| < |a_2| = |a_3| < |a_4|$. Under these circumstances, a flip will happen on the second and third dimensions of $\mathbf{a}$.

$$S_{soft}(\mathbf{a}) = [0, 0, 0, \text{sign}(a_4)(|a_4| - |a_2|)]$$

Without loss of generality we assume $|a_1'| < |a_2'| \leq |a_3'| < |a_4'|$. Thus,

$$S_{soft}(\mathbf{a}') = [0, 0, \text{sign}(a_3')(|a_3'| - |a_2'|), \text{sign}(a_4')(|a_4'| - |a_2'|)].$$

The proof of the fourth dimension is similar to 1), so we only focus on $a_3$.

$$\text{sign}(a_3')(|a_3'| - |a_2'|)$$
$$\leq |\,|a_3'| - |a_2'|\,|$$
$$\leq |\,|a_3'| - |a_3|\,| + |\,|a_2'| + |a_2|\,|$$
$$\leq 2\delta$$

Take $\delta \leq \frac{1}{2}\epsilon$ and this is done.

3) If there exists three or four equivalent in $\mathbf{a}$, a flip will happen at the second and third dimension. Thus, these cases can be reduced to 1) or 2).

$\square$

## A.2  GLUE scores of GPT-2

See Table 9.

## A.3  Acceleration

See Table 10.

Table 9: Comparison between GLUE scores of different pre-train methods on GPT-2 models. This table is the elaboration of Table 6.

| Params | Method | Avg score | CoLA | MNLI | MRPC | QNLI | QQP | RTE | SST-2 | STS-B | WNLI |
|---|---|---|---|---|---|---|---|---|---|---|---|
| 124M | Dense | 73.9 ± 1.1 | 44.6 ± 0.9 | 82.0 ± 0.1 | 78.3 ± 1.3/84.8 ± 1.0 | 88.4 ± 0.2 | 90.0 ± 0.0 | 86.5 ± 0.0/61.3 ± 1.5 | 91.9 ± 0.2 | 77.3 ± 3.2/77.9 ± 2.9 | 24.3 ± 7.1 |
| | T-SR-STE+DF [20] | 74.3 ± 0.5 | 44.8 ± 1.3 | 81.5 ± 0.2 | 77.5 ± 1.8/84.2 ± 1.3 | 87.8 ± 0.1 | 89.5 ± 0.1 | 85.9 ± 0.1/66.0 ± 1.0 | 90.6 ± 0.4 | 80.0 ± 0.8/80.3 ± 0.5 | 23.9 ± 6.4 |
| | T-SR-STE | 72.6 ± 0.2 | 41.9 ± 0.3 | 81.0 ± 0.2 | 76.3 ± 0.9/83.4 ± 0.7 | 87.0 ± 0.3 | 89.3 ± 0.1 | 85.6 ± 0.1/60.6 ± 3.4 | 90.9 ± 0.4 | 76.2 ± 3.2/76.5 ± 3.0 | 21.8 ± 4.4 |
| | SR-STE | 73.8 ± 0.3 | 38.3 ± 2.8 | 80.9 ± 0.2 | 79.7 ± 0.7/85.9 ± 0.6 | 87.1 ± 0.3 | 89.5 ± 0.1 | 85.8 ± 0.2/65.5 ± 1.0 | 90.5 ± 0.4 | 80.9 ± 1.2/80.9 ± 1.1 | 20.1 ± 1.8 |
| | **S-STE** | **74.1 ± 0.4** | **42.3 ± 1.1** | **80.5 ± 2.8** | **79.3 ± 1.9/85.6 ± 1.4** | **88.1 ± 0.2** | **89.8 ± 0.1** | **86.2 ± 0.1/62.9 ± 1.1** | **91.9 ± 0.4** | **81.0 ± 1.1/81.2 ± 1.0** | **20.8 ± 3.6** |
| 350M | Dense | 76.3 ± 0.1 | 54.3 ± 0.4 | 85.1 ± 0.1 | 80.7 ± 1.0/86.6 ± 0.7 | 90.7 ± 0.1 | 91.0 ± 0.1 | 87.8 ± 0.1/64.9 ± 1.7 | 93.5 ± 0.4 | 81.7 ± 1.2/82.2 ± 0.8 | 17.6 ± 3.2 |
| | T-SR-STE+DF [20] | 77.1 ± 0.2 | 51.8 ± 1.8 | 84.3 ± 0.1 | 80.6 ± 1.3/86.5 ± 0.8 | 90.4 ± 0.2 | 90.7 ± 0.1 | 87.5 ± 0.1/66.7 ± 1.3 | 93.3 ± 0.4 | 83.4 ± 1.1/83.5 ± 1.1 | 26.4 ± 4.0 |
| | T-SR-STE | 76.3 ± 0.4 | 50.0 ± 1.7 | 84.1 ± 0.2 | 81.4 ± 1.5/87.1 ± 1.1 | 90.0 ± 0.3 | 90.6 ± 0.1 | 87.3 ± 0.1/67.9 ± 1.5 | 93.3 ± 0.4 | 81.3 ± 1.5/81.4 ± 1.4 | 20.6 ± 3.8 |
| | SR-STE | 76.8 ± 0.4 | 47.2 ± 3.0 | 84.3 ± 0.2 | 81.4 ± 0.9/87.2 ± 0.6 | 90.2 ± 0.1 | 90.8 ± 0.1 | 87.6 ± 0.1/68.3 ± 1.4 | 93.9 ± 0.1 | 82.0 ± 1.6/82.0 ± 1.7 | 27.1 ± 3.1 |
| | **S-STE** | **76.9 ± 0.6** | **54.2 ± 1.7** | **84.6 ± 0.2** | **80.2 ± 1.3/86.1 ± 0.9** | **90.5 ± 0.3** | **90.8 ± 0.1** | **87.5 ± 0.2/65.1 ± 1.9** | **93.7 ± 0.4** | **83.6 ± 1.1/83.8 ± 1.1** | **22.5 ± 3.9** |
| 774M | Dense | 76.2 ± 0.4 | 57.5 ± 2.0 | 86.1 ± 0.1 | 80.3 ± 1.3/86.4 ± 0.9 | 91.4 ± 0.2 | 91.1 ± 0.1 | 88.0 ± 0.1/67.7 ± 2.6 | 94.6 ± 0.4 | 77.3 ± 3.3/78.4 ± 2.9 | 15.1 ± 2.3 |
| | T-SR-STE+DF [20] | 77.1 ± 0.4 | 55.9 ± 0.9 | 85.6 ± 0.2 | 81.2 ± 0.6/87.0 ± 0.4 | 91.4 ± 0.1 | 91.0 ± 0.1 | 87.8 ± 0.1/71.5 ± 0.7 | 94.2 ± 0.4 | 81.8 ± 1.3/82.3 ± 1.2 | 15.8 ± 1.2 |
| | **S-STE** | **78.6 ± 0.8** | **57.3 ± 2.7** | **86.6 ± 0.2** | **80.6 ± 1.4/86.6 ± 0.9** | **92.0 ± 0.1** | **91.5 ± 0.1** | **88.5 ± 0.1/78.3 ± 1.5** | **94.9 ± 0.3** | **85.5 ± 1.2/85.7 ± 1.1** | **16.1 ± 5.9** |

Table 10: Pre-training acceleration ratio with different different batch size $N$, sequence length $n$, embedding dimension $d$ and heads number $h$ on single FFN block and transformer block of GPT-2 with RTX 3090 GPUs.

| | N | n | d | h | FFN | GPT-2 |
|---|---|---|---|---|---|---|
| Pre-train | 4 | 2048 | 5120 | 40 | 1.31 | 1.18 |
| | 16 | 2048 | 7168 | 56 | 1.32 | 1.18 |
| | 8 | 2048 | 7168 | 56 | 1.33 | 1.17 |
| | 4 | 2048 | 7168 | 56 | 1.31 | 1.17 |
| | 4 | 2048 | 9216 | 72 | 1.31 | 1.18 |
| Inference | 16 | 2048 | 7168 | 56 | 1.54 | 1.23 |
| | 8 | 2048 | 7168 | 56 | 1.46 | 1.15 |

## A.4 Limitations

As we propose accuracy results of S-STE on several tasks, no actual acceleration result is given. While theoretically 2x faster results can be expected (FP8 quantization), the NVIDIA acceleration library (cuSPARSElt [31]) is not satisfactory, which causes inconvenience on implementation. The peak FLOPS of 2:4-spMM is lower than theoretical GEMM FLOPS; see Table 11.

Table 11: Peak FLOPS of general matrix multiplications (GEMMs) and 2:4 sparse matrix multiplications (2:4-spMMs) on H100. The size we take to test is $16384 \times 16384 \times 16384$.

| | GPU | FP8 Tensor Core |
|---|---|---|
| Specifications | H100 PCIe 2:4-spMM | 3200 TFLOPS |
| | H100 PCIe GEMM | 1600 TFLOPS |
| | H100 SXM 2:4-spMM | 4000 TFLOPS |
| | H100 SXM GEMM | 2000 TFLOPS |
| Actual results with cuSPARSElt | H100 SXM 2:4-spMM | 1900 TFLOPS |
| | H100 SXM GEMM | 1500 TFLOPS |

Table 12: GPU Hours of pre-training models on RTX 4090.

| | GPU Hours |
|---|---|
| GPT-2 124M | 400 |
| GPT-2 350M | 900 |
| GPT-2 774M | 2500 |
| Transformer-base | 30 |
| DeiT-base | 120 |

### A.5 Broader Impact

S-STE can be used mainly to accelerate the pre-training stage of large-scale networks, like LLaMA [43] and GPT-4 [33]. Theoretically GEMMs of the FFN layer can be accelerated up to 4x faster than FP16 dense models, which would greatly reduce the electric power consumption of pre-training modern large-scale models. However, this method may also used for some models that is non-compliance with regulations and ethics, such as models that generate discriminatory contents.

### A.6 Experiments compute resources

To replicate our experiments, we provide the estimated GPU hours of each setting; see Table 12.

