# OpenReview forum: "S-STE: Continuous Pruning Function for Efficient 2:4 Sparse Pre-training"
_NeurIPS.cc/2024/Conference — NeurIPS 2024 poster_

### Official Review · Reviewer_WZj6 · 2024-07-08

**Soundness:** 2
**Presentation:** 2
**Contribution:** 2
**Rating:** 4
**Confidence:** 4

**Summary:**

The paper discusses three drawbacks of traditional N:M sparse training and suggests using soft-thresholding over hard-thresholding for 2:4 sparse pre-training. It introduces the idea of rescaling sparse weights with a fixed scaling factor per tensor. Results from experiments in machine translation, image classification, and large generative language models demonstrate its effectiveness.

**Strengths:**

1.	The idea is simple and clear. The proposed approach is very easy to understand and implement.

2.	The paper discusses several important issues for N:M sparse pre-training. These discussions are very useful for practice.

3.	The experiments verify the selection of some important parameters, and also show the effectiveness of the proposed approach on several tasks.

**Weaknesses:**

1.	Most of the techniques are discussed in other tasks, and the paper is only applied to N:M sparse pre-training, which is not innovative enough. So the contributions are not enough for NeurIPS.

2.	It is very important to discuss the drawbacks of existing N:M sparse pre-training, but these discussions have not led to a new algorithm. There is insufficient inevitability between these discussions and the proposed algorithm.

3.	The experiments are weak. There is a lack of important comparison methods and real training acceleration.

**Questions:**

I have the following comments, which may improve the quality of the paper.

1.	The discussion of formula (4) should use very rigorous mathematical derivation, however, the discussion of related contents is very imprecise.

2.	In Theorem 4.1, it is better to provide the definition of continuous projection for a vector. Additionally, it is important to address why hard-thresholding and other algorithms fail to meet this property, and experimentally demonstrate the performance improvement that can result from adhering to this property.

3.	In Table 1, is the \gamma choice dependent on the \beta setting? Does it establish the beta before selecting \gamma?

4.	What is the reason for simulating dense weights? Why perform fixed weight rescaling \beta?

5.	In Table 4, it is better to separate “dense” and use it as a baseline. The best result in the second column is marked incorrectly.

6.	There are many methods for N:M sparse pre-training as shown in the Related Work section. The paper needs to make a detailed comparison with these methods.

7.	The core of this paper is the pre-training acceleration. However, it is far from enough to discuss the possible acceleration in theory, and it is necessary to carry out sufficient experiments to compare the acceleration effects of various methods.

8.	All the figures in the paper are too small to see clearly.

9.	Most references lack the names of conferences or journals.

**Limitations:**

Not applicable.

---

> ### Author Rebuttal · Authors · 2024-08-05
>
> Dear Reviewer WZj6,
>
> Thank you for the acknowledgment of the potential and effectiveness of our work and the detailed constructive comments. Below we provide a point-to-point response to all comments.
>
> **Weakness 1:** Most of the techniques are discussed in other tasks, and the paper is only applied to N:M sparse pre-training, which is not innovative enough. So the contributions are not enough for NeurIPS.
>
> **Reply:** We argue that the contributions are adequate:
>
> 1. Although most of the techniques we discuss is rooted in prior work, we propose to **innovatively implementing soft-thresholding on 2:4 pre-training tasks**. Besides, we make a couple of **highly original 2:4 specific modifications** to utilize those methods (choosing optimal threshold, freezing scaling factor, minimizing MSE).  These are the **technological contributions** of our study.
> 2. More importantly, we are trying to **push the frontiers of relatively unpopular 2:4 pre-train research**: on the one hand, as pre-training is very difficult, 2:4 pre-training is even more tough because we need to **jointly optimize activated masks and their weight** values. In this way, **our work reveal the difficulty of the problem**. On the other hand, prior work on 2:4 pre-training topic mostly focus on optimization skills, however **they never consider continuity as a main problem**. Our study tries to get to the essence of the question and we provide a deeper understanding of it. We **not only set up new SOTA baselines for future study to follow, but also provide an important and new research insight for future work to explore**.
>
> **Weakness 2:** It is very important to discuss the drawbacks of existing N:M sparse pre-training, but these discussions have not led to a new algorithm. There is insufficient inevitability between these discussions and the proposed algorithm.
>
> **Reply:** We believe that inevitability is quite clear:
>
> 1. The most important drawback we see from previous work is discontinuity. To elaborate this, we observe three phenomena:  incorrect descending direction, inability to predict the amount of descent and oscillation (Sec. 3).
> 1. To overcome the drawback of previous pruning function, we introduce 2:4 specific soft-thresholding pruning function to S-STE and theoretically prove its continuity.
> 1. To complete the circle, we show that **S-STE successfully overcome the discontinuity phenomena mentioned above** via experiment; see Sec. 4, Fig. 2(c)(d) and 4(d).

---

> > ### Comment · Reviewer_WZj6 · 2024-08-11
> > **Thanks for your hard work. I will raise my score by 1, but not higher.**
> >
> > I appreciate the authors' detailed response and will raise my score accordingly. Nevertheless, I still believe that using soft-thresholding as a surrogate function for hard-thresholding is a common practice, which somewhat diminishes the paper's innovation. Additionally, the discussed weaknesses of the existing methods pertain to post-analysis, which does not contribute to the development of new methods.

---

> > > ### Author Response · Authors · 2024-08-11
> > > **Thnak you! Please remember to update the score ☺**
> > >
> > > Thank you for your time and effort in reviewing our work and discussion! Please do remember to update the score in the openreview system as we haven't seen changes yet. Thank you so much!

---

> ### Author Response · Authors · 2024-08-05
> **Response to Reviewer WZj6 (cont.)**
>
> **Weakness 3:** The experiments are weak. There is a lack of important comparison methods and real training acceleration.
>
> **Reply:** About comparison baselines:
>
> Please refer to Question 6.
>
> About acceleration:
>
> To further respond to your doubts about real-world acceleration, we report end-to-end acceleration as follows. For experiment setting, we choose different sizes of GPT-2 models and test acceleration with FP16 weights. **For inference, we achieve 1.53x speedup with FFN layer and 1.23x speedup with the network; for pre-training, we achieve 1.32x speedup for FFN layer and 1.18x speedup for the network.** This will be updated in our paper.
>
>
>
> *Table.* Pre-training acceleration ratio with different different batch size $N$, sequence length $n$, embedding dimension $d$ and heads number $h$ on GPT-2 with RTX 3090 GPUs.
>
> |N|n|d|h|acceleration@FFN|acceleration@GPT-2|
> |-|-|-|-|-|-|
> |4|2048|5120|40|1.309071284|1.176265882|
> |16|2048|7168|56|1.317673412|1.18020863|
> |8|2048|7168|56|1.325846831|1.173059355|
> |4|2048|7168|56|1.308463658|1.171455338|
> |4|2048|9216|72|1.311344165|1.176620318|
>
> *Table.* Inference acceleration ratio with different different batch size $N$, sequence length $n$, embedding dimension $d$ and heads number $h$ on GPT-2 with RTX 3090 GPUs.
>
> |N|n|d|h|acceleration@FFN|acceleration@GPT-2|
> |-|-|-|-|-|-|
> |16|2048|7168|56|1.536392435|1.233632|
> |8|2048|7168|56|1.464448312|1.149633|
>
>
>
> To further investigate how we reach \~1.2x speedup, we profile our code and break down the time costs as shown in the table below.
>
>
>
> *Table.* Time costs of each part of our network and the dense model in one iteration per layer. $m$ denotes the accumulation steps over micro batches. Our method is evaluated on GPT-2, with batch size 16, sequence length 1024, embedding dimension 1024 and heads number 16.
>
> |||||dense (ms/exec)|sparse (ms/exec)|acceleration ratio|frequency (exec/iter)|
> |-|-|-|-|-|-|-|-|
> |ffn|linear|fwd|GEMM|12173.8|7305.8|1.67|1|
> |||bwd|GEMM|23295|18688|1.25|1|
> ||||mvue+prune|0|171.4|-|1|
> ||||total|23295|18859.4|1.63|1|
> |||**total**||**35468.8**|**21558**|**1.24**|1|
> ||others [17]|fwd||167|118.2|-|1|
> |||bwd||65.5|20|-|1|
> |||total||232.5|138.2|-|1|
> ||total|fwd||12340.8|7424|1.66|1|
> |||bwd||23360.5|18879.4|1.24|1|
> |||total||35701.3|26303.4|1.36|1|
> |others [18]||fwd||6874.3|7090.6|-|1|
> |||bwd||13920.7|14117.5|-|1|
> |||total||20795|21208|-|1|
> |total||fwd||19215.1|14514.5|1.32|1|
> |||bwd||37281.2|32,996.9|1.13|1|
> |||**total**||**56496.3**|**47511.4**|**1.19**|1|
> |prune weight||||0|320.3|-|$\frac{1}{m}$|
>
>
>
> Due to the reason that 1) previous work [1,13] achieves similar acceleration ratio on the same settings and 2) we only accelerate two matrix multiplications for pre-training linear layer while previous work [1] accelerate all three multiplications, we believe that the acceleration is reasonable.
>
> Based on the results above, we believe the overheads of continuous weight pruning function is negligible. According to the time cost table above, the cost of continuous weight pruning function per iteration is
> $$
> 320.3 \times \frac{1}{m} = \frac{320.3}{m} ms.
> $$
> Compared to other parts ($47511.4ms$ for the whole iteration), this is indeed negligible.
>
> It is worth noting that the acceleration we achieve is made on RTX 3090 GPUs with FP16 data type. As we try our best to achieve real acceleration effect on H100 GPUs with popular FP8 precision, the acceleration test fail because FP8 2:4-spMM don't even meet dense baseline; see table below. We are in contact with NVIDIA to address this issue and hopefully will get reasonable results in the future.
>
> *Table.* Peak FLOPS of general matrix multiplications (GEMMs) and 2:4 sparse matrix multiplications (2:4-spMMs) on H100.
>
> ||GPU|FP8 Tensor Core|
> |-|-|-|
> |Specifications|H100 PCIE 2:4-spMM|3200 TFLOPS|
> ||H100 PCIE GEMM|1600 TFLOPS|
> ||H100 SXM 2:4-spMM|4000 TFLOPS|
> ||H100 SXM GEMM|2000 TFLOPS|
> |Actual results with cuSPARSElt|H100 SXM 2:4-spMM|1900 TFLOPS|
> ||H100 SXM GEMM|1500 TFLOPS|

---

> ### Author Response · Authors · 2024-08-05
> **Response to Reviewer WZj6 (cont.)**
>
> **Question 1:** The discussion of formula (4) should use very rigorous mathematical derivation, however, the discussion of related contents is very imprecise.
>
> **Reply:** We apologize that a typo exists and elaborate this as follows.
>
> In formula (4), we consider a dense model with the simplest batch gradient method, which is defined by
> $$
> {\\mathbf{w}}\_{k+1} = {\\mathbf{w}}\_{k}-\\alpha\_k \\nabla\_{{\\mathbf{w}}\_k} F({\\mathbf{w}}\_k)
> $$
> iteratively.
>
> According to Taylor's formula, we have
> $$
> F({\\mathbf{w}}\_{k+1}) = F({\\mathbf{w}}\_{k}) + \\nabla\_{{\\mathbf{w}}\_k} F({\\mathbf{w}}\_k)^\\top({\\mathbf{w}}\_{k+1}-{\\mathbf{w}}\_{k})+o(||{\\mathbf{w}}\_{k+1}-{\\mathbf{w}}\_{k}||)
> $$
> Combined with the batch gradient method formula, we have
> $$
> \\begin{align}
> F({\\mathbf{w}}\_{k+1}) - F({\\mathbf{w}}\_{k})
> &= \\nabla\_{{\\mathbf{w}}\_k} F({\\mathbf{w}}\_k)^\\top({\\mathbf{w}}\_{k+1}-{\\mathbf{w}}\_{k})+o(||{\\mathbf{w}}\_{k+1}-{\\mathbf{w}}\_{k}||) \\\\
> &= -\\alpha\_k\\nabla\_{{\\mathbf{w}}\_k} F({\\mathbf{w}}\_k)^\\top \\nabla\_{{\\mathbf{w}}\_k} F({\\mathbf{w}}\_k)+o(||{\\mathbf{w}}\_{k+1}-{\\mathbf{w}}\_{k}||) \\\\
> &\\approx -\\alpha\_k||\\nabla\_{{\\mathbf{w}}\_k} F({\\mathbf{w}}\_k)||^2
> \\end{align}
> $$
> **Question 2:** In Theorem 4.1, it is better to provide the definition of continuous projection for a vector.
>
> **Reply:** The definition of continuous projection for a vector:
>
> 1. This definition is similar to the definition of continuity of a normal function, i.e. $g:\mathbb{R}^n \to\mathbb{R}^n$ is continuous at $\mathbf{w}_0$, when:
>
>    $\forall \epsilon>0, \exists \delta>0$, s.t. when $||\mathbf{w}-\mathbf{w}_0||<\delta$, $||g(\mathbf{w})-g(\mathbf{w}_0)||<\epsilon$.
>
> 2. This can also be defined as the function is continuous for every output variable, based on continuity of multivariable function:
>
>    $g:\mathbb{R}^n \to\mathbb{R}^n$ is continuous when all $f_i(\mathbf{w}) = (g(\mathbf{w}))_{[i]}$ is continuous for $i=1,2,...,n,$ where $f_i(\mathbf{w})$ is the $i$-th output element of $g$.
>
> **Question 3:** Additionally, it is important to address why hard-thresholding and other algorithms fail to meet this property, and experimentally demonstrate the performance improvement that can result from adhering to this property.
>
> **Reply:** Why hard-thresholding and other algorithms fail to meet this property:
>
> 1. For hard-thresholding-based methods like STEP [9], the pruning function is obviously discontinuous when a "flip" happens; please refer to Fig. 3 for a intuitive explanation. (In Fig. 3, that hard-thresholding is discontinuous when $|a_1|=|a_2|$.) This can be directly observed and understood, needless of mathematical proof.
>
> 2. For SR-STE [1,8] method, it adds an extra decay term to improve accuracy: $
>    \min_{\mathbf{w}} F(\mathbf{\tilde w})+\tfrac{\lambda_W}{2} \Vert {\mathbf{w}} \odot \overline{m({\mathbf{w}})}\Vert_2^2;$ see Sec. 3.4. The SR-STE [8] authors show its effectiveness by experiment, but they fail to realize that the decay term works as a "smoother". In this way, they only partially smooth the function, and the effect is limited: the function is rigorously continuous only when $\lambda_W \rightarrow \infty$, which is impossible practically. This is the main drawback of SR-STE's continuity.
>
> 3. Compared to those methods, S-STE is a completely continuous pruning function and has the advantage over hard-thresholding-based methods and SR-STE-based methods. All experiments we make in Sec. 6 demonstrates that, with the above baselines compared. Please refer to Table 4,5,6,8 for more details.
>
>    We further add a few more experiments to address your skepticism.
>
>    *Table.* Experimental results for DeiT.
>
>    |Size|Method|Acc@1|Acc@5|
>    |-|-|-|-|
>    |DeiT-tiny|Original|72.2|91.1|
>    ||SR-STE|67.8| 88.6 |
>    ||**S-STE**|**68.5**|**88.9** |
>    |DeiT-small|Original|79.9|94.5|
>    || SR-STE [10]|75.7|-|
>    || Bi-Mask [10]|77.6|-|
>    ||**S-STE**| **78.5**|**94.4**|
>
>    *Table.* Different fine-tuning results on GLUE and SQuAD.
>
>    |Model|Downstream task|Pre-training method|Fine-tuning method|Avg score|
>    |-|-|-|-|-|
>    |GPT-2 124M|GLUE|S-STE|S-STE|$74.1\pm0.4$|
>    |GPT-2 124M|GLUE|S-STE|hard-thresholding|$73.9\pm0.6$|
>    |GPT-2 124M|SQuAD|S-STE|S-STE|$68/78.8$|
>    |GPT-2 124M|SQuAD|S-STE|hard-thresholding|$67.6/78.6$|

---

> ### Author Response · Authors · 2024-08-05
> **Response to Reviewer WZj6 (cont.)**
>
> **Question 4:** In Table 1, is the $\gamma$ choice dependent on the $\beta$ setting? Does it establish the $\beta$ before selecting $\gamma$?
>
> **Reply:** No. Choosing threshold and setting weight rescaling are totally independent. In the control experiment in Table 1, $\beta$ is set according to the algorithm proposed in Sec. 4.2.
>
> **Question 5:** What is the reason for simulating dense weights?
>
> **Reply:** The reason for simulating dense weights:
>
> 1. Since the network is designed as a dense network originally, activation magnitude and variance changes if no rescaling is applied. The best way to recover this loss is to compare the sparse model with dense equivalent and scale back weights compared to the dense model. Other techniques like dropout and the original soft-thresholding paper [???] have the similar weight rescaling step, and both of those methods are comparing the sparse weights (activations) with the dense ones.
>
> 2. To further prove the effectiveness of rescaling sparse weights to the dense magnitude, more ablation study is made. Results show that the effect of weight rescaling is not obvious on computer vision tasks like DeiT-small, but is significant for language models like Transformer-base. The reason behind this is due to the difference of their tasks: classification tasks are usually easier to complete than generation tasks, and the change may not be well reflected in the accuracy on simpler tasks.
>
>    *Table.* Experimental result of S-STE (soft-thresholding and weight rescaling), MVUE and FP8 training with DeiT-small on ImageNet-1K.
>
>    |soft-thresholding|weight rescaling|$\operatorname{MVUE}(\nabla_{\mathbf{Z}}^\top)$|FP8|comment|test acc1|test acc5|
>    |-|-|-|-|-|-|-|
>    |-|-|×|×|dense|79.9|95|
>    |-|-|×|√|dense; FP8|79.7|94.9|
>    |×|×|×|×|hard-thresholding|77.7|93.9|
>    |√|√|×|×||78.8|94.6|
>    |√|×|×|×||78.9|94.7|
>    |√|√|×|√||78.6|94.4|
>    |√|√|√|×||78.9|94.6|
>    |√|×|√|×||78.2|94.2|
>    |**√**|**√**|**√**|**√**||**78.5**|**94.4**|
>
>    Besides, we'd like to kindly point out that another control experiment done on Transformer-base with WMT 14 En-De has already been presented; see Table 3 in the paper. To further clarify this, we expand this table to another ablation study, which presents the results with Transformer-base settings; see table below.
>
>    *Table.* Experimental result of S-STE (soft-thresholding and weight rescaling), MVUE and FP8 training with Transformer-base on WMT 14 En-De.
>
>    | soft-thresholding|weight rescaling|$\operatorname{MVUE}(\nabla_{\mathbf{Z}}^\top)$ | FP8 | comment | test BLEU | validation loss | average epoch loss |
>    |-|-|-|-|-|-|-|-|
>    |-|-|×|×|dense|26.42|3.977| 4.555 |
>    |×|×|×|×|hard-thresholding|25.65|4.088 |4.686 |
>    |√|×|×|×||25.28|4.044|4.67 |
>    |√|√|×|×||26.3|4.007|4.605 |
>    |√|√|√|×||25.93|4.01|4.602 |
>    |**√**|**√**|**√**|**√**||**26.11**|**4.011**|**4.61**|

---

> ### Author Response · Authors · 2024-08-05
> **Response to Reviewer WZj6 (cont.)**
>
> **Question 6:** Why perform fixed weight rescaling $\beta$?
>
> **Reply:** The reason to perform fixed weight rescaling but not dynamic weight rescaling:
>
> 1. We admit that related work such as original soft-thresholding paper [14] perform weight rescaling dynamically, which means $\beta$ is computed on every forward propagation.
> 1. However, intuitively, for 2:4 specific soft-thresholding, it is different. Original dense weight never participate in forward calculation. Because S-STE needs to subtract third largest weight in a group of four, what matters are the relative values of weights, not the absolute value. Thus, rescaling after or during training is meaningless.
> 1. Further experiments confirm our suspicions. As explained in our paper, dynamic $\beta$ results in extremely large $\beta$ at the last few layers, and consequently results in large flip rate. This makes training unstable and injures accuracy. All the experiments and analysis are in Sec. 4.2.
>
> **Question 7:** In Table 4, it is better to separate “dense” and use it as a baseline. The best result in the second column is marked incorrectly.
>
> **Reply:** "Dense" here is actually used as a baseline. We will modify the format of this table to present information clearly, thank you!
>
> **Question 8:** There are many methods for N:M sparse pre-training as shown in the Related Work section. The paper needs to make a detailed comparison with these methods.
>
> **Reply:** It's true that there are a lot of traditional dynamic sparse training methods and post-training pruning methods, but those methods have different mission backgrounds and purposes from ours, and is meaningless to be used as contrast. The main experimental setting we choose has **already covered the full spectrum of 2:4 related work**. Here is a detailed summarization as follows.
>
> 1. Typical soft-pruning methods. In "2:4 pre-training" context, we are talking about **practically accelerating pre-training via 2:4 property** [1]. In the light of this purpose, traditional soft-pruning methods like Decaying Pruning [6] which can't possibly accelerate training is meaningless to be considered. Besides, since our sparsity is fixed to 50%, comparing with different sparsity or dynamic sparsity methods may be unfair as it's a pre-training task. Typical works include Decaying Pruning.
> 2. Other dynamic sparse training methods. Traditional dynamic sparse training methods such as RigL [2], Early-Bird Tickets [3,4] can't acquire acceleration as well. Because of this, those methods are not our competitors.
> 3. Post-training methods. We are doing a pre-training task. Post-training methods such as  Deep Compression [7], lottery ticket [15], Sanity-Checking [5] and even 2:4 one-shot pruning [16] which has nothing to do with pre-training shouldn't be considered too.
> 4. 2:4 pre-training methods (weight sparsity). They are the main baselines we shall compare with. They include SR-STE [8] and its improvement [1], STEP [9] and Bi-Mask [10].
> 5. Other 2:4 training methods. Other 2:4 pre-training works, such as MVUE [11] and T-mask [12] focus on different questions and is also meaningless to be used as contrast.
>
> **Question 9:** The core of this paper is the pre-training acceleration. However, it is far from enough to discuss the possible acceleration in theory, and it is necessary to carry out sufficient experiments to compare the acceleration effects of various methods.
>
> **Reply:** That's a very good question because in 2:4 pre-training related work, very few have reported real acceleration. Most of the work stay on simulation and report only accuracy. In all 2:4 pre-training works, only paper [1] has reported real acceleration and we mention this in Weakness 2. Besides, PyTorch [12] reported acceleration on BERT inference and can be used as a reference too.
>
> **Question 10:** All the figures in the paper are too small to see clearly.
>
> **Reply:** We realize this problem and we will redraw most of the illustrations that are too small. The previous compression of format is due to insufficient layout.
>
> **Question 11:** Most references lack the names of conferences or journals.
>
> **Reply:** We apologize for the oversight and would like to thank you for pointing them out. We will check all our references and correct those with non-standard citations. For the papers that appear in conferences or journals, we will replace the arXiv links with the publisher links.

---

> ### Author Response · Authors · 2024-08-05
> **Response to Reviewer WZj6 (cont.)**
>
> [1] Accelerating Transformer Pre-training with 2:4 Sparsity, https://proceedings.mlr.press/v235/hu24r.html
>
> [2] Rigging the Lottery: Making All Tickets Winners, https://arxiv.org/abs/1911.11134
>
> [3] Drawing Early-Bird Tickets: Towards More Efficient Training of Deep Networks, https://arxiv.org/abs/1909.11957
>
> [4] EarlyBERT: Efficient BERT Training via Early-bird Lottery Tickets, https://arxiv.org/abs/2101.00063
>
> [5] Sanity-Checking Pruning Methods: Random Tickets can Win the Jackpot, https://arxiv.org/abs/2009.11094
>
> [6] Training Recipe for N:M Structured Sparsity with Decaying Pruning Mask, https://arxiv.org/abs/2209.07617
>
> [7] Compressing Deep Neural Networks with Pruning, Trained Quantization and Huffman Coding, https://arxiv.org/abs/1510.00149
>
> [8] Learning N:M Fine-grained Structured Sparse Neural Networks From Scratch, https://arxiv.org/abs/2102.04010
>
> [9] STEP: Learning N:M Structured Sparsity Masks from Scratch with Precondition, https://arxiv.org/abs/2302.01172
>
> [10] Bi-directional Masks for Efficient N:M Sparse Training, https://arxiv.org/abs/2302.06058
>
> [11] Minimum Variance Unbiased N:M Sparsity for the Neural Gradients, https://arxiv.org/abs/2203.10991
>
> [12] Accelerated Sparse Neural Training: A Provable and Efficient Method to Find N:M Transposable Masks, https://arxiv.org/abs/2102.08124
>
> [13] (prototype) Accelerating BERT with semi-structured (2:4) sparsity, https://pytorch.org/tutorials/prototype/semi_structured_sparse.html
>
> [14] Are Straight-Through gradients and Soft-Thresholding all you need for Sparse Training?, https://arxiv.org/abs/2212.01076
>
> [15] The Lottery Ticket Hypothesis: Finding Sparse, Trainable Neural Networks, https://arxiv.org/abs/1803.03635
>
> [16] Accelerating Sparse Deep Neural Networks, https://arxiv.org/abs/2104.08378
>
> [17] All functions in FFN except linear layers, i.e. activation function and dropout.
>
> [18] All other parts in the network except FFN layers, e.g. attention, optimizer, etc.

---

### Official Review · Reviewer_73MA · 2024-07-11

**Soundness:** 3
**Presentation:** 3
**Contribution:** 2
**Rating:** 4
**Confidence:** 3

**Summary:**

This paper presents a framework to circumvent the common challenges associated with STE-based 2:4 pre-training due to pruning function discontinuity. In particular, their framework addresses 3 aspects of this behavior: descent direction, amount of descent, and sparse mask oscillation.

**Strengths:**

* This paper provides a comprehensive analysis of the limitations of traditional pruning techniques, with a particular emphasis on how their framework addresses these challenges. I believe there is a novelty in the exploration of continuous pruning strategies.
* Additionally, each component is comprehensively validated – for example, the problem of mask oscillation seems to be adequately documented in Section 3.3 with a viable exemplar to demonstrate impact.

**Weaknesses:**

* This work is centered on the potential of continuous pruning schemes for enabling faster sparse pre-training. However, I do see a strong resemblance with conventional soft mask approaches to pruning schemes. In particular, I wonder if the authors considered typical soft-pruning works in their comparisons, and if not, what was the reasoning for that experimental setting choice.
* I found the experimental results to be quite sparse in model selection and competitive method benchmarking. Predominantly this method benchmarks against SR-STE and the dense framework, however, there isn’t enough context as to why other pruning-based methods are not included in scope. Further, there seem to be a few different tasks ablated however comprehensive details on why each application and/or model was selected are missing. For example, if demonstrating the efficacy of a sparse pre-training method it, would be beneficial to see the scaling effect on large models, say ViT-B/L for ImageNet-1K or the SWIN architectures.

**Questions:**

Please address the questions in the weaknesses section.

**Limitations:**

/

---

> ### Author Rebuttal · Authors · 2024-08-04
>
> Dear Reviewer 73MA,
>
> Thank you for the acknowledgment of the potential and effectiveness of our work and the detailed constructive comments. Below we provide a point-by-point response to all comments.
>
> **Weakness 1:** This work is centered on the potential of continuous pruning schemes for enabling faster sparse pre-training. However, I do see a strong resemblance with conventional soft mask approaches to pruning schemes. In particular, I wonder if the authors considered typical soft-pruning works in their comparisons, and if not, what was the reasoning for that experimental setting choice.
>
> **Reply:** We are not considering typical soft-pruning methods as our comparisons. These methods have different mission backgrounds and purposes from ours, and is meaningless to be used as contrast.
>
> In "2:4 pre-training" context, we are talking about practically accelerating pre-training via 2:4 property [1]. In the light of this purpose, traditional dynamic sparse training methods such as RigL [2], Early-Bird Tickets [3,4], Sanity-Checking [5], Decaying Pruning [6] which can't possibly accelerate training shouldn't be considered. Besides, post-training methods such as  Deep Compression [7] which has nothing to do with pre-training phase shouldn't be considered too. The main experimental setting we choose has already covered the full spectrum of 2:4 related work, including hard-thresholding [8], SR-STE [8], STEP [9] and Bi-Mask [10]. Other 2:4 pre-training works, such as MVUE [11] and T-mask [12] focus on different questions and is also meaningless to be used as contrast.
>
> **Weakness 2:** I found the experimental results to be quite sparse in model selection and competitive method benchmarking. Predominantly this method benchmarks against SR-STE and the dense framework, however, there isn’t enough context as to why other pruning-based methods are not included in scope.
>
> **Reply:** For model selection, there are three reasons to explain why we choose these settings.
>
> 1. **We are solving a very difficult task** and pre-training is quite consuming for academic research group. Different from post-training research which can leverage state-of-the-art model architectures like Llama 3, Phi-3 and Qwen2, pre-training where we **train a model from scratch** takes lots of hardware resources and thousands of GPU hours. Thus the model sizes are likely to be smaller than most of the other post-training pruning studies.
>
> 2. As pre-training is very difficult, 2:4 pre-training is even more tough because we need to **jointly optimize activated masks and their weight** values. **The difficulty of this problem leads to insufficient baseline works from the past.** That's the reason we focus on this task; **we not only reveal the difficulty of the task, but also set up new SOTA baselines**. In other words, we are trying to **push the frontiers of relatively unpopular 2:4 pre-train research**.
>
> 3. The models we use are chosen from a wide range of most representative works in deep learning with the acceptable size and available settings, spanning from natural language processing to computer vision tasks. For NLP tasks, we choose classical Transformer-base and GPT-2; for CV tasks, we choose popular Vision Transformer. We believe **these representative works can set up new SOTA baselines for future study to follow**.
>
> For competitive method benchmarking and other pruning-based methods, please refer to Weakness 1.
>
> **Weakness 3:** Further, there seem to be a few different tasks ablated however comprehensive details on why each application and/or model was selected are missing. For example, if demonstrating the efficacy of a sparse pre-training method it, would be beneficial to see the scaling effect on large models, say ViT-B/L for ImageNet-1K or the SWIN architectures.
>
> **Reply:** For model selection, please refer to Weakness 2 and Weakness 1. For the scaling effect of large models, we pre-train different scales of GPT-2 and Vision Transformer. We list the result as follows and hopefully it solves your doubts. Basically, the performance get closer to or surpass dense baseline as model size increases.
>
>
>
> *Table.* Experimental results for DeiT. DeiT-base fail to due to infinity loss on SR-STE and S-STE.
>
> |Size|Method|Acc@1|Acc@5|
> |-|-|-|-|
> | DeiT-tiny |Original|72.2|91.1|
> ||SR-STE|67.8|88.6|
> ||**S-STE**| **68.5** |**88.9**|
> |DeiT-small| Original |79.9|94.5|
> ||SR-STE [10]|75.7|-|
> ||Bi-Mask [10]|77.6|-|
> ||**S-STE**| **78.5** |**94.4**|
>
> *Table.* SQuAD scores of different sizes and pre-training methods on GPT-2. Similar to Table 5, we use 2:4 sparse weights to evaluate S-STE model, while dense parameters to evaluate the rest. Note that the results in Table 8 are close to the dense baselines, which means the room to improve is already small. Other experiments in Table 5 can show a larger improvement than SR-STE.
>
> |Params|Pre-training method|Fine-tuning method|EM|F1|
> |-|-|-|-|-|
> |124M|Dense| Dense|67.6|78.8|
> ||Transposable SR-STE+Dense|Dense|67.5|78.5|
> ||SR-STE|Dense|66.2|77.5|
> ||**S-STE**| **S-STE**|**68**|**78.8**|
> |350M|Dense| Dense | 73.2 | 83.6 |
> ||Transposable SR-STE+Dense|Dense|71.9|82.4|
> ||SR-STE|Dense|72.0|82.4|
> ||**S-STE**|**S-STE**|**72.2**|**82.7** |
> |774M|Dense|Dense|74.3|84.9|
> ||Transposable SR-STE+Dense|Dense|74.3|84.6|
> ||**S-STE**|**S-STE**|**75.5**|**85.5**|

---

> ### Author Response · Authors · 2024-08-04
> **Response to Reviewer 73MA (cont.)**
>
> [1] Accelerating Transformer Pre-training with 2:4 Sparsity, https://proceedings.mlr.press/v235/hu24r.html
>
> [2] Rigging the Lottery: Making All Tickets Winners, https://arxiv.org/abs/1911.11134
>
> [3] Drawing Early-Bird Tickets: Towards More Efficient Training of Deep Networks, https://arxiv.org/abs/1909.11957
>
> [4] EarlyBERT: Efficient BERT Training via Early-bird Lottery Tickets, https://arxiv.org/abs/2101.00063
>
> [5] Sanity-Checking Pruning Methods: Random Tickets can Win the Jackpot, https://arxiv.org/abs/2009.11094
>
> [6] Training Recipe for N:M Structured Sparsity with Decaying Pruning Mask, https://arxiv.org/abs/2209.07617
>
> [7] Compressing Deep Neural Networks with Pruning, Trained Quantization and Huffman Coding, https://arxiv.org/abs/1510.00149
>
> [8] Learning N:M Fine-grained Structured Sparse Neural Networks From Scratch, https://arxiv.org/abs/2102.04010
>
> [9] STEP: Learning N:M Structured Sparsity Masks from Scratch with Precondition, https://arxiv.org/abs/2302.01172
>
> [10] Bi-directional Masks for Efficient N:M Sparse Training, https://arxiv.org/abs/2302.06058
>
> [11] Minimum Variance Unbiased N:M Sparsity for the Neural Gradients, https://arxiv.org/abs/2203.10991
>
> [12] Accelerated Sparse Neural Training: A Provable and Efficient Method to Find N:M Transposable Masks, https://arxiv.org/abs/2102.08124

---

> ### Author Response · Authors · 2024-08-12
> **Sincerely looking forward to further discussions**
>
> Dear Reviewer 73MA,
>
> We hope that our response and revision have adequately addressed your concerns. If our efforts have indeed addressed your concerns, we would be very grateful if you could reconsider our work and possibly adjust the score accordingly. If you have any additional questions or suggestions, we would be happy to have further discussions.
>
> Best regards,
>
> The Authors

---

### Official Review · Reviewer_JQrs · 2024-07-13

**Soundness:** 4
**Presentation:** 4
**Contribution:** 4
**Rating:** 8
**Confidence:** 5

**Summary:**

The authors address the challenge of pre-training models, with 2:4 sparsity, to high quality (ideally matching a densely-trained model).  Building on existing methods, the authors point out three issues with discontinuous pruning functions: gradients move in the wrong direction, weight updates do not match expectation, and values can oscillate between being masked and not-masked.  Demonstrating these shortcomings and analyzing the source leads the authors to propose two modifications to the traditional straight-through estimator approach: 2:4-specific soft thresholding and fixed weight rescaling.  The former provides a continuous pruning function, and the latter compensates for the reduced weight magnitude caused by the soft thresholding.  Targeted experiments show that the proposed method does not suffer from the three shortcomings of traditional approaches.  In full training experiments, the authors also show that applying MVUE (an existing technique) to data gradients and using FP8 representations do not interfere with S-STE's success.  Machine translation, image classification, and generative tasks are used for testing various models with 2:4 sparsity applied to the MLPs of transformer blocks and the data indicates that the resulting models are superior to baselines, and in several cases comparable in quality to the dense baselines.

**Strengths:**

**Originality**
While the individual advancements, soft-thresholding and weight rescaling, are rooted in prior work, I think the combination and the modifications made by the authors are highly original.

**Quality**
The experiments performed are precisely what are needed, and no more.  Each claim is supported with one clear figure or table so that the reader immediately sees that the authors considered the various angles and have shown why the angle they have selected is the best.  The breadth of full training experiments shows that the proposed method applies to image classification, machine translation, and text generation with particular success in the latter two.  Further, they combined their method with two largely orthogonal compression techniques: FP8 data representations for training and sparsifying another tensor (data gradients).  This makes the results even more compelling.

**Clarity**
I found the submissions to be easy to read in general and with an appropriate amount of detailing previous and related work.  The few issues I had were minor and listed below, and they did not detract from the well-organized paper.

**Significance**
This work is of high importance.  It has done a great deal to close the gap between dense and sparse training for a constrained type of sparsity that offers practical acceleration in readily available hardware.  If it continues to be successful in broader experiments in the wild, then it could be widely adopted, saving significant resources in training new networks and reduce the latency of performing large-scale experiments.

**Weaknesses:**

**Originality**
N/A

**Quality**
A missing experiment is one that would show the relative importance of scaling factor *beta*.  It would be a good addition to the ablation study presented in Table 7.

**Clarity**

Table 2 was confusing at first.  It wasn't obvious that the first row, without S-STE, was just dense training, rather than a different DST baseline.  Once I understood this, its low loss value not being the "winner" made sense.  The second confusing bit was that the second row had the next-lowest loss, but was also not **emphasized**.  Again, I realized that this is because it would not lead to acceleration in the backwards pass.  This second point could be clarified by adding "... and accelerates the backwards pass" to the caption.

In line 256, the authors say they "choose to sparsify del(Z) only in the backwards pass," which sounds like in the forwards pass, they do not sparsify del(Z).  Clearly, they don't, because this term is not involved in the forwards pass, but it caught me off guard - I think "choose to sparsify only del(Z) in the backwards pass," as opposed to sparsifying both del(Z) and S(W), is more clear.

In line 331, I think the authors mean to say that FP8 quantization accelerates GEMMs up to 2x faster than their 16b counterparts.  (As detailed in Section 5.2, the upper bound of speedup from dense BF16/FP16 to sparse FP8 is indeed 4x.)

I noticed a few typos (there may be more lurking):

- Line 13: "our method surpass" -> surpasses

- Line 174: "contains two main partitions" -> parts

- Line 201: "the closer *t* is from |*t3*|" -> "the closer *t* is to |*t3*|"

- Line 263: "forwrad" -> "forward"

- Line 326: "leaverage" -> "leverage"

**Significance**
Transformer blocks typically have two more weighted linear layers that were left dense in this work: the QKV projection before attention and the attention output projection.  Thus, there is potentially more memory and computation savings available, but without experiments, it is unknown if model quality will remain high if they were also made sparse.

**Questions:**

- If a lower flip rate is better in Figure 4, is it concerning that SR-STE's flip rate is lower than S-STE's?  (If not, why not?)

- Would S-STE apply to all N:M configurations by subtracting the N+1th largest entry in soft-thresholding, or is it truly 2:4- (and, given Figure 3, 1:2-) specific?

**Limitations:**

The discussion of the limitations in the appendix is appreciated, but it seems to suggest that sparsity should give a theoretical 4x increase in throughput, but I believe this should be 2x.  (FP8 can theoretically give another 2x.)  Also, the sizes of the GEMMs that gave these results should be listed.  Finally, I'd point out that the method has only been tested on two out of four linear layers in Transformer networks, not other types of networks, including covolutional networks.

---

> ### Author Rebuttal · Authors · 2024-08-04
>
> Dear Reviewer JQrs,
>
> Thank you for the acknowledgment of the potential and effectiveness of our work and the detailed constructive comments. Below we provide a point-to-point response to all comments.
>
> **Weakness 1:** **Quality** A missing experiment is one that would show the relative importance of scaling factor *beta*. It would be a good addition to the ablation study presented in Table 7.
>
> **Reply:** We appreciate this insightful suggestion and add content to the ablation study as below. We add more experiments and redraw the ablation table. Results show that the effect of weight rescaling is not obvious on computer vision tasks like DeiT-small, but is significant for language models like Transformer-base. The reason behind this is due to the difference of their tasks: classification tasks are usually easier to complete than generation tasks, and the change may not be well reflected in the accuracy on simpler tasks.
>
> *Table.* Experimental result of S-STE (soft-thresholding and weight rescaling), MVUE and FP8 training with DeiT-small on ImageNet-1K.
>
> |soft-thresholding|weight rescaling|$\operatorname{MVUE}(\nabla_{\mathbf{Z}}^\top)$|FP8|comment|test acc1|test acc5|
> |-|-|-|-|-|-|-|
> |-|-|×|×|dense|79.9|95|
> |-|-|×|√|dense; FP8|79.7|94.9|
> |×|×|×|×|hard-thresholding|77.7|93.9|
> |√|√|×|×||78.8|94.6|
> |√|×|×|×||78.9|94.7|
> |√|√|×|√||78.6|94.4|
> |√|√|√|×||78.9|94.6|
> |√|×|√|×||78.2|94.2|
> |**√**|**√**|**√**|**√**||**78.5**|**94.4**|
>
> Besides, we'd like to kindly point out that another control experiment done on Transformer-base with WMT 14 En-De has already been presented; see Table 3 in the paper. To further clarify this, we expand this table to another ablation study, which presents the results with Transformer-base settings; see table below.
>
> *Table.* Experimental result of S-STE (soft-thresholding and weight rescaling), MVUE and FP8 training with Transformer-base on WMT 14 En-De.
>
> | soft-thresholding|weight rescaling|$\operatorname{MVUE}(\nabla_{\mathbf{Z}}^\top)$ | FP8 | comment | test BLEU | validation loss | average epoch loss |
> |-|-|-|-|-|-|-|-|
> |-|-|×|×|dense|26.42|3.977| 4.555 |
> |×|×|×|×|hard-thresholding|25.65|4.088 |4.686 |
> |√|×|×|×||25.28|4.044|4.67 |
> |√|√|×|×||26.3|4.007|4.605 |
> |√|√|√|×||25.93|4.01|4.602 |
> |**√**|**√**|**√**|**√**||**26.11**|**4.011**|**4.61**|
>
>
>
> The new ablation study as well as the supplemental experimental data of the first one will be updated in our paper.
>
> **Weakness 2:** **Clarity** Table 2 was confusing at first. It wasn't obvious that the first row, without S-STE, was just dense training, rather than a different DST baseline. Once I understood this, its low loss value not being the "winner" made sense. The second confusing bit was that the second row had the next-lowest loss, but was also not **emphasized**. Again, I realized that this is because it would not lead to acceleration in the backwards pass. This second point could be clarified by adding "... and accelerates the backwards pass" to the caption.
>
> **Reply:** We realize the problem and redraw the table as follows. Besides, we will consider putting Table 2 beside Sec. 5, where it should have been. The previous deviation of format is due to insufficient layout.
>
> *Table.* Results of different MVUE strategies on GPT-2 774M with 4000 steps. Sparsifying $S(\mathbf{W})^\top$ and $\nabla_{\mathbf{Z}}^\top$ can accelerate the two matrix multiplications of backward pass respectively. However, accuracy loss introduced by those sparse matrices are different.
>
> | S-STE | $\operatorname{MVUE}(S(\mathbf{W})^\top)$ | $\operatorname{MVUE}(\nabla_{\mathbf{Z}}^\top)$ | comment | loss |
> |-|-|-|-|-|
> |-|×|×|dense|3.3948|
> |-|×|×|SR-STE|3.4739|
> |√|×|×||3.4333|
> |√|√|×||3.4644|
> |√|√|√||3.4773|
> |√|×|√||**3.448**|
>
> **Weakness 3:** **Clarity** In line 256, the authors say they "choose to sparsify del(Z) only in the backwards pass," which sounds like in the forwards pass, they do not sparsify del(Z). Clearly, they don't, because this term is not involved in the forwards pass, but it caught me off guard - I think "choose to sparsify only del(Z) in the backwards pass," as opposed to sparsifying both del(Z) and S(W), is more clear.
>
> **Reply:** The latter expression is definitely less confusing. Thanks for reminding us! We will update this in our paper.
>
> **Weakness 4:** **Clarity** In line 331, I think the authors mean to say that FP8 quantization accelerates GEMMs up to 2x faster than their 16b counterparts. (As detailed in Section 5.2, the upper bound of speedup from dense BF16/FP16 to sparse FP8 is indeed 4x.)
>
> **Reply:** Yes, that's exactly what we mean. In the updated version of our paper, this line will be replaced with
>
> > While 16-bit float tensors are widely used in pre-training, FP8 – where float numbers stored in 8 bits – is a popular quantization methods which theoretically accelerates GEMMs up to 4x faster than its fp32 counterparts and 2x faster than its FP16/BF16 counterparts.
>
> **Weakness 5:** **Typos** I noticed a few typos (there may be more lurking).
>
> **Reply:** We apologize for the oversight and would like to thank you for pointing them out. All the clerical errors that you mention and some other typos are corrected in the updated version.
>
> **Weakness 6:** **Significance** Transformer blocks typically have two more weighted linear layers that were left dense in this work: the QKV projection before attention and the attention output projection. Thus, there is potentially more memory and computation savings available, but without experiments, it is unknown if model quality will remain high if they were also made sparse.
>
> **Reply:** That's a very good question. We have done experiments with the four QKV projection layers, and the output is not satisfying. Sparsifying QKV projections may require attention specific methods, designed in conjunction with attention mechanism. We decide to leave them to future work.

---

> ### Author Response · Authors · 2024-08-04
> **Response to Reviewer JQrs (cont.)**
>
> **Question 1:** If a lower flip rate is better in Figure 4, is it concerning that SR-STE's flip rate is lower than S-STE's? (If not, why not?)
>
> **Reply:** No, there's no concern on flip rate of S-STE. The comparison on flip rate between SR-STE and S-STE successfully shows the advantage of S-STE. We clarify this theory in three steps.
>
> 1) Reiterate **the principle of flip rate**. As pointed out by [1], the core of a healthy 2:4 training is to have the flip rate first rises then decreases in the training process. In other words, **neither lower nor higher flip rate is better. Different training stages requires the flip rate to act differently; the "peak" of flip rate curve should be high enough and the "tail" should be low enough**. The reason behind this, as they point out, is to make the network first explore and set up connection modes then frozen the connections and fine-tune weights, which aligns with the model's behavior proposed by [2].
>
> 2) Convert this ambiguous principle into clear guidelines. In practice, in order to apply this principle, we usually consider the dense model to be the flip rate standard. The closer a method gets to the standard curve, the better performance it will get. Flip rate curves that deviate from the standard is considered poisonous.
>
> 3) Observe flip rate results in SR-STE and S-STE. As shown in Figure 4(d) of the paper, flip rate curve of dense model and SR-STE doesn't well coincide, which means there's drawbacks in SR-STE training. (To be specific, **the "peak" of SR-STE is not high enough**.) On the other hand, flip rate of S-STE and dense nearly overlaps with each other, which is a perfect match under our assumptions.
>
> Your misunderstanding might come from the fact that 1) larger $\beta$ results in higher flip rate; 2) dynamically computing $\beta$ leads to extremely large $\beta$ and extremely higher flip rate, which is harmful. These facts are both true but they don't collide with the principle above. If a "tail" is extremely higher than a standard one, the result shouldn't be optimistic. But if a "peak" is lower than the standard peak, the result can suffer from accuracy loss as well.
>
> **Question 2:** Would S-STE apply to all N:M configurations by subtracting the N+1th largest entry in soft-thresholding, or is it truly 2:4- (and, given Figure 3, 1:2-) specific?
>
> **Reply:** **S-STE is compatible with all N:M patterns** for the reason that **its continuity is guaranteed theoretically**. We limit the study only on 2:4 (and 1:2) is due to hardware bounds nowadays. In future research, when more N:M patterns can be explored, S-STE can still be  applied by subtracting the N+1th largest entry.
>
> **Limitation 1:** The discussion of the limitations in the appendix is appreciated, but it seems to suggest that sparsity should give a theoretical 4x increase in throughput, but I believe this should be 2x. (FP8 can theoretically give another 2x.)
>
> **Reply:** The 4x theoretical increase is because that we take FP8 into account, but we now realize that this will lead to confusions. We will clarify this in the same way as Weakness 4.
>
> **Limitation 2:** Also, the sizes of the GEMMs that gave these results should be listed.
>
> **Reply:** The size of GEMMs and 2:4-spMMs we take is $16384\times16384\times16384$. This will be updated in our paper.
>
> **Limitation 3:** Finally, I'd point out that the method has only been tested on two out of four linear layers in Transformer networks, not other types of networks, including convolutional networks.
>
> **Reply:** The main goal of our work targets on transformers. This is because traditional 2:4 training methods like SR-STE [3], STEP [4] and Bi-Mask [5] have already achieved very good performance on convolutional networks, but none of those methods perform well on transformers. We believe that it's not very meaningful to continue focus on conv networks, but it's valuable to develop new transformer specific 2:4 pre-training strategies. As for other architectures like Mamba and KAN, 2:4 training algorithms needs to be designed in conjunction with architecture, which is our future work.
>
>
>
> [1] Accelerating Transformer Pre-training with 2:4 Sparsity, https://proceedings.mlr.press/v235/hu24r.html
>
> [2] Drawing Early-Bird Tickets: Towards More Efficient Training of Deep Networks, https://arxiv.org/abs/1909.11957
>
> [3] Learning N:M Fine-grained Structured Sparse Neural Networks From Scratch, https://arxiv.org/abs/2102.04010
>
> [4] STEP: Learning N:M Structured Sparsity Masks from Scratch with Precondition, https://arxiv.org/abs/2302.01172
>
> [5] Bi-directional Masks for Efficient N:M Sparse Training, https://arxiv.org/abs/2302.06058
>
> [6] Mamba: Linear-Time Sequence Modeling with Selective State Spaces, https://arxiv.org/abs/2312.00752
>
> [7] KAN: Kolmogorov-Arnold Networks, https://arxiv.org/abs/2404.19756

---

> > ### Comment · Reviewer_JQrs · 2024-08-09
> > **Thank you for the detailed responses**
> >
> > I appreciate the detailed responses to my questions and concerns.  I'll keep my Strong Accept rating and urge my fellow reviewers to revisit their scores given all your responses.

---

> ### Author Response · Authors · 2024-08-10
> **Thank you!**
>
> Thank you so much for your Strong Accept! We believe the theoretical and experimental progress in 2:4 sparse training can bring new solutions for large transformers' pre-training and inference acceleration. Thank you!

---

### Official Review · Reviewer_18hX · 2024-07-14

**Soundness:** 3
**Presentation:** 3
**Contribution:** 3
**Rating:** 5
**Confidence:** 4

**Summary:**

This work studies an efficient method for pre-training, identifying three significant limitations in previous 2:4 sparse pre-training approaches: incorrect descent direction, the inability to predict the extent of descent, and oscillations in the sparse mask. Subsequently, the authors introduce a novel training methodology that integrates a continuous weight projection technique with a rescaling strategy, thereby enhancing pre-training efficiency and achieving performance comparable to conventional full-parameter training.

**Strengths:**

1. The structure of this work is well organized. It begins by highlighting the current challenges in 2:4 sparse pre-training, where the suboptimal performance largely stems from the discontinuity of the pruning function. This issue is then explored through various examples, followed by a demonstration of the proposed methods.
2. The experiments conducted encompass a variety of model sizes and datasets.
3. The motivation behind this work is well-founded and clearly articulated.

**Weaknesses:**

- The acceleration of S-STE is demonstrated solely in terms of theoretical gains. Seems like S-STE will introduce extra computation cost, which might sacrify part of the acceleration. Studying the end-to-end acceleration would further enhance the quality of this work.
- Is the discontinuity issue discussed in Section 3 associated with the learning rate? A larger learning rate results in more substantial weight updates, potentially causing more frequent flipping. Additionally, is this issue still a big problem during fine-tuning scenarios, where the learning rate is generally lower?
- The improvements of S-STE are modest, e.g., in Table 8, showing a 0.3 improvement in the F1 score for GPT-2 models with 124M and 350M parameters.

**Questions:**

Please refer to the weakness.

**Limitations:**

There is no end-to-end acceleration results evaluated of current methods.

---

> ### Author Rebuttal · Authors · 2024-08-04
>
> Dear Reviewer 18hX,
>
> Thank you for recognizing the potential and effectiveness of our work and for providing detailed constructive comments. Below, we address each point raised.
>
> **Weakness 1:** The acceleration of S-STE is demonstrated solely in terms of theoretical gains. Seems like S-STE will introduce extra computation cost, which might sacrify part of the acceleration. Studying the end-to-end acceleration would further enhance the quality of this work.
>
> **Reply:** It it true that S-STE will introduce extra computation cost, but be argue the impact won't be significant:
>
> 1. In a FFN block, the computational complexity of a matrix multiplication with shape $M\times N\times K$ is $O(MNK)$. Other operations like activation function, pruning function and weight rescaling are considered **element-wise**, with computation cost bounded to $O(MN)$ or $O(NK)$. Compared to the $O(MNK)$ parts of a FFN block, this cost is negligible.
> 2. As micro-batching technique is applied most of the time, the cost of the pruning and weight rescaling operation will be further reduced to $\frac{1}{m}$, where $m$ denotes the accumulation steps over micro batches.
>
> To further respond to your doubts about real-world acceleration, we report end-to-end acceleration as follows. For experiment setting, we choose different sizes of GPT-2 models and test acceleration with FP16 weights. **For inference, we achieve 1.53x speedup with FFN layer and 1.23x speedup with the network; for pre-training, we achieve 1.32x speedup for FFN layer and 1.18x speedup for the network.** This will be updated in our paper.
>
>
>
> *Table.* Pre-training acceleration ratio with different different batch size $N$, sequence length $n$, embedding dimension $d$ and heads number $h$ on GPT-2 with RTX 3090 GPUs.
>
> |N|n|d|h|acceleration@FFN|acceleration@GPT-2|
> |-|-|-|-|-|-|
> |4|2048|5120|40|1.309071284|1.176265882|
> |16|2048|7168|56|1.317673412|1.18020863|
> |8|2048|7168|56|1.325846831|1.173059355|
> |4|2048|7168|56|1.308463658|1.171455338|
> |4|2048|9216|72|1.311344165|1.176620318|
>
> *Table.* Inference acceleration ratio with different different batch size $N$, sequence length $n$, embedding dimension $d$ and heads number $h$ on GPT-2 with RTX 3090 GPUs.
>
> |N|n|d|h|acceleration@FFN|acceleration@GPT-2|
> |-|-|-|-|-|-|
> |16|2048|7168|56|1.536392435|1.233632|
> |8|2048|7168|56|1.464448312|1.149633|
>
>
>
> To further investigate how we reach \~1.2x speedup, we profile our code and break down the time costs as shown in the table below.
>
>
>
> *Table.* Time costs of each part of our network and the dense model in one iteration per layer. $m$ denotes the accumulation steps over micro batches. Our method is evaluated on GPT-2, with batch size 16, sequence length 1024, embedding dimension 1024 and heads number 16.
>
> |||||dense (ms/exec)|sparse (ms/exec)|acceleration ratio|frequency (exec/iter)|
> |-|-|-|-|-|-|-|-|
> |ffn|linear|fwd|GEMM|12173.8|7305.8|1.67|1|
> |||bwd|GEMM|23295|18688|1.25|1|
> ||||mvue+prune|0|171.4|-|1|
> ||||total|23295|18859.4|1.63|1|
> |||**total**||**35468.8**|**21558**|**1.24**|1|
> ||others [2]|fwd||167|118.2|-|1|
> |||bwd||65.5|20|-|1|
> |||total||232.5|138.2|-|1|
> ||total|fwd||12340.8|7424|1.66|1|
> |||bwd||23360.5|18879.4|1.24|1|
> |||total||35701.3|26303.4|1.36|1|
> |others [3]||fwd||6874.3|7090.6|-|1|
> |||bwd||13920.7|14117.5|-|1|
> |||total||20795|21208|-|1|
> |total||fwd||19215.1|14514.5|1.32|1|
> |||bwd||37281.2|32,996.9|1.13|1|
> |||**total**||**56496.3**|**47511.4**|**1.19**|1|
> |prune weight||||0|320.3|-|$\frac{1}{m}$|
>
>
>
> Due to the reason that 1) previous work [1] achieves similar acceleration ratio on the same settings and 2) we only accelerate two matrix multiplications for pre-training linear layer while previous work [1] accelerate all three multiplications, we believe that the acceleration is reasonable.
>
> Based on the results above, we believe the overheads of continuous weight pruning function is negligible. According to the time cost table above, the cost of continuous weight pruning function per iteration is
> $$
> 320.3 \times \frac{1}{m} = \frac{320.3}{m} ms.
> $$
> Compared to other parts ($47511.4ms$ for the whole iteration), this is indeed negligible.
>
> It is worth noting that the acceleration we achieve is made on RTX 3090 GPUs with FP16 data type. As we try our best to achieve real acceleration effect on H100 GPUs with popular FP8 precision, the acceleration test fail because FP8 2:4-spMM don't even meet dense baseline; see table below. We are in contact with NVIDIA to address this issue and hopefully will get reasonable results in the future.
>
> *Table.* Peak FLOPS of general matrix multiplications (GEMMs) and 2:4 sparse matrix multiplications (2:4-spMMs) on H100.
>
> ||GPU|FP8 Tensor Core|
> |-|-|-|
> |Specifications|H100 PCIE 2:4-spMM|3200 TFLOPS|
> ||H100 PCIE GEMM|1600 TFLOPS|
> ||H100 SXM 2:4-spMM|4000 TFLOPS|
> ||H100 SXM GEMM|2000 TFLOPS|
> |Actual results with cuSPARSElt|H100 SXM 2:4-spMM|1900 TFLOPS|
> ||H100 SXM GEMM|1500 TFLOPS|

---

> ### Author Response · Authors · 2024-08-04
> **Response to Reviewer 18hX (cont.)**
>
> **Weakness 2:** Is the discontinuity issue discussed in Section 3 associated with the learning rate? A larger learning rate results in more substantial weight updates, potentially causing more frequent flipping.
>
> **Reply:** Learning rate is part of the reason for the discontinuity issue, but it's not the main factor. **The main factor is built-in discontinuity of loss function (and pruning function).** Let's explain it in two ways.
>
> > [!NOTE]
> >
> > Here we use flip rate to represent the stability of training, and in a sense to represent the continuity of the loss function; see Sec. 3.3. Higher flip rate means more flips are taken in an optimizer step, and suggests that the network is more discrete; lower flip rate means the weight is changing smoothly, which is the signal of a more continuous function.
>
>
>
> *Table.* Flip rate of different methods on different time steps with Transformer-base and WMT En-De dataset. Note that dense, S-STE and hard-thresholding methods share the same learning rate, while "hard-thresholding2" halve the learning rate with rest of the conditions remain the same.
>
> |Step|Dense|S-STE|hard-thresholding|hard-thresholding2|
> |-|-|-|-|-|
> |5k|0.001852|0.001883|0.002487|0.001476|
> |20k|8.51e-4|8.83e-4|0.001879|0.001488|
> |40k|5.25e-4|5.62e-4|0.001789|0.001466|
> |60k|3.92e-4|4.22e-4|0.001731|0.001459|
> |80k|3.43e-4|3.69e-4|0.001824|0.001547|
> |100k|3.11e-4|3.47e-4|0.001868|0.001674|
> |Final test BLEU|26.42|26.3|25.38|24.99|
>
>
>
> First, **within a single pre-training procedure**, flip rate change is not necessarily related to learning rate. As shown in table above, the flip rate of hard-thresholding is extremely large even when learning rate is close to zero at the end of pre-training. This denotes that **small learning rate not necessarily slow down flipping, and decreasing learning rate doesn't relieve the discontinuity problem**.
>
> Second, **comparison between two hard-thresholding pre-training processes** show that even if we halve learning rate, flip rate would not necessarily halve as well; rather it may remain extremely high at the end of training. This denotes that **choosing a small learning rate will not effectively decrease flip rate**.
>
> **Weakness 3:** Additionally, is this issue still a big problem during fine-tuning scenarios, where the learning rate is generally lower?
>
> **Reply:** Yes, the problem still exists.
>
> 1. According to Weakness 2, low learning rate doesn't imply the optimization target to be continuous and smooth and smooth.
>
> 2. In fine-tuning scenarios, weights are changing slightly. This is more like the later phase of pre-training, where learning rate is close to zero and flip rate is low. Previous study [1] points out that high flip rate (i.e. discontinuity) affects performance even more severe on later phase of pre-training and fine-tuning occasions than early phase of pre-training.
>
> 3. We further prove this via a control experiment; see table below. Although the model's ability is mostly related to pre-train methods, different fine-tuning methods do show slight differences.
>
>
>
>    *Table.* Different fine-tuning results on GLUE and SQuAD.
>
>    |Model|Downstream task|Pre-training method|Fine-tuning method|Avg score|
>    |-|-|-|-|-|
>    |GPT-2 124M|GLUE|S-STE|S-STE|$74.1\pm0.4$|
>    |GPT-2 124M|GLUE|S-STE|hard-thresholding|$73.9\pm0.6$|
>    |GPT-2 124M|SQuAD|S-STE|S-STE|$68/78.8$|
>    |GPT-2 124M|SQuAD|S-STE|hard-thresholding|$67.6/78.6$|

---

> ### Author Response · Authors · 2024-08-04
> **Response to Reviewer 18hX (cont.)**
>
> **Weakness 4:** The improvements of S-STE are modest, e.g., in Table 8, showing a 0.3 improvement in the F1 score for GPT-2 models with 124M and 350M parameters.
>
> **Reply:** We apologize that this is mainly because the description of the experiment settings are unclear. We now clarify this with more experimental results and a new table.
>
> *Table.* SQuAD scores of different sizes and pre-training methods on GPT-2. Similar to Table 5, we use 2:4 sparse weights to evaluate S-STE model, while dense parameters to evaluate the rest. Note that the results in Table 8 are close to the dense baselines, which means the room to improve is already small. Other experiments in Table 5 can show a larger improvement than SR-STE.
>
> |Params|Pre-training method|Fine-tuning method|EM|F1|
> |-|-|-|-|-|
> |124M|Dense|Dense|67.6|78.8|
> ||TransposableSR-STE+Dense|Dense|67.5|78.5|
> ||SR-STE|Dense|66.2|77.5|
> ||**S-STE**|**S-STE**|**68**|**78.8**|
> |350M|Dense|Dense|73.2|83.6|
> ||TransposableSR-STE+Dense|Dense|71.9|82.4|
> ||SR-STE|Dense|72.0|82.4|
> ||**S-STE**|**S-STE**|**72.2**|**82.7**|
> |774M|Dense|Dense|74.3|84.9|
> ||Transposable SR-STE+Dense|Dense|74.3|84.6|
> ||**S-STE**|**S-STE**|**75.5**|**85.5**|
>
> According to the detailed result, we argue the advantage of S-STE is significant:
>
> 1. The SR-STE baseline is **already close to dense baseline**, which means that each small performance improvement is significant.
> 2. In the original paper, the SR-STE baseline contains a dense stage in pre-training, which helps to recover accuracy. However, **S-STE can match and surpass the improvement without the dense stage**.
> 3. We forget to mention that for other baseline, we all use dense fine-tuning, which means **S-STE competes with full parameter models with only half of the FFN parameters**. This advantage is very significant.
> 4. The **main goal** to conduct downstream task here is **to compare the different pre-train methods**. Now that SQuAD scores are close, **we can refer to other downstream tasks like GLUE**. On these tasks, **S-STE also achieves non-negligible improvement compared to baselines; see Table 5 in paper**.
>
> **Limitation 1:** There is no end-to-end acceleration results evaluated of current methods.
>
> **Reply:** Please refer to Weakness 1.
>
>
>
> [1] Accelerating Transformer Pre-training with 2:4 Sparsity, https://proceedings.mlr.press/v235/hu24r.html
>
> [2] All functions in FFN except linear layers, i.e. activation function and dropout.
>
> [3] All other parts in the network except FFN layers, e.g. attention, optimizer, etc.

---

> ### Author Response · Authors · 2024-08-12
> **Sincerely looking forward to further discussions**
>
> Dear Reviewer 18hX,
>
> We hope that our response and revision have adequately addressed your concerns. If our efforts have indeed addressed your concerns, we would be very grateful if you could reconsider our work and possibly adjust the score accordingly. If you have any additional questions or suggestions, we would be happy to have further discussions.
>
> Best regards,
>
> The Authors

---

### Decision · Program_Chairs · 2024-09-25

**Decision:**

Accept (poster)

**Comment:**

Almost all of the reviewers agreed that the work was well-motivated, well-written and had significant novelty. Reviewer's concerns primarily centred on the lack of real-world timings for the proposed method, increased computational requirements of the proposed S-STE method, the modest gains in accuracy shown, lack of ablation over the scaling hyperparameter $\beta$, a similarity with soft-mask approaches for pruning, and a lack of comparison with pruning methods.

Overall it appears that many of these concerns appear to have been addressed in the rebuttal by the authors. The post-rebuttal discussion amongst all the reviewers (except one) was also productive and further clarified I believe that that the most significant of each reviewer's concerns were addressed by the authors. As such, with the feedback from the reviewers/rebuttal incorporated, I believe this paper will be a relevant and significant contribution.